# CONTEXT-PARAMETRIC INVERSION: WHY INSTRUCTION FINETUNING CAN WORSEN CONTEXT RELIANCE

**Sachin Goyal**[*†]  **Christina Baek**[*†]   **J. Zico Kolter**[†]   **Aditi Raghunathan**[†]
Carnegie Mellon University[†]
{sachingo,kbaek,zkolter,raditi}@cs.cmu.edu

## ABSTRACT

A standard practice when using large language models is for users to supplement their instruction with an input context containing new information for the model to process. However, models struggle to reliably follow the input context, especially when it conflicts with their parametric knowledge from pretraining. In-principle, one would expect models to adapt to the user context better after instruction finetuning, particularly when handling knowledge conflicts. However, we observe a surprising failure mode: during instruction tuning, the context reliance under knowledge conflicts initially increases as expected, but then *gradually decreases as instruction finetuning progresses*. This happens while the performance on standard benchmarks keeps on increasing far after this drop. We call this phenomenon **context-parametric inversion** and observe it across multiple general purpose instruction tuning datasets such as TULU, Alpaca and Ultrachat, across different model families like Llama, Mistral, and Pythia. We perform various controlled studies and theoretical analysis to show that context-parametric inversion occurs due to examples in the instruction finetuning data where the input context provides information that aligns with model's parametric knowledge. Our analysis suggests some natural mitigation strategies with limited but insightful gains, and serves as a useful starting point in addressing this deficiency in instruction finetuning.

## 1 INTRODUCTION

Large language models (LLMs) are widely used for a variety of tasks, many of which require carefully balancing the knowledge embedded in their parameters with the information provided through the input context. A persistent challenge, however, is their tendency to overrely on parametric knowledge, even when it contradicts with the context. This overreliance hinders the ability to update model facts with augmented contexts and reliably follow atypical user instructions (Qiu et al., 2023; Adlakha et al., 2024). This tension between contextual and parametric knowledge has been commonly studied under the moniker of *knowledge conflicts*. Existing works explore various decoding and finetuning remedies (Shi et al., 2023; Yuan et al., 2024; Longpre et al., 2022; Chen et al., 2022), but model behavior under knowledge conflicts remain difficult to control, and conflicts often occur more frequently scale (McKenzie et al., 2024). Moreover, we have limited understanding of the underlying dynamics that drive models to ignore the context and rely heavily on its parametric knowledge.

In this work, we study the effect of instruction finetuning (IFT)—a staple part of the LLM pipeline—on the ability to override pretrained knowledge through the context. IFT seeks to enhance the model's ability to assist with user queries. Oftentimes, these instructions contains a context with critical information needed to complete the task. For instance, an instruction "`What is the total price of my trip to Hawaii?`" operates on a context "`Context: [Itinerary List]`", and an instruction "`Rank these famous soccer players based on these scores`" could contain a context like: "`[Scores Table].`" In these circumstances, instruction tuned models must appropriately leverage the input context to respond, instead of relying on parametric knowledge. However, we make an intriguing observation during IFT, where in the presence of knowledge conflicts, *the model's reliance on context initially increases as expected but surprisingly starts decreasing.*

---

[*]Equal Contribution.

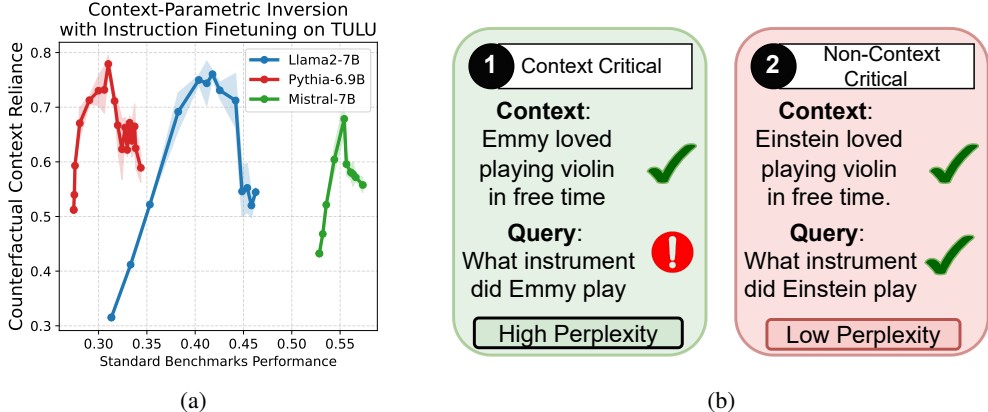

(a)  (b)

Figure 1: (a) **Context-Parametric Inversion** In the presence of knowledge conflicts, context reliance first increases and then decreases during the process of instruction finetuning. (b) Instruction datasets often include both context-critical examples and non-context-critical examples. This latter group effectively causes the decline in context reliance (§ 4.3).

We measure the context reliance by designing inputs contexts that suggest a fictional answer to a user query different from facts in the pretraining corpus (§ 3.2). We evaluate context reliance across the IFT trajectory of multiple instruction datasets —TULU, Alpaca or UltraChat — and multiple model families — Llama, Pythia and Mistral. Across these settings, we see that context reliance initially increases and then decreases, a phenomenon we call **context-parametric inversion**. In fact, this drop begins in early timesteps of IFT, while the performance on standard benchmarks (e.g., MMLU, GSM8k, SQuAD) keeps on increasing far after this drop. For example, as shown in Figure 1a, the context reliance of Llama2-7B (as measured on knowledge conflict datasets (§ 3.2)) increases from 30% to 60% initially with IFT. However, it start dropping as the finetuning progresses further, dipping to around 35%.

Why do we observe context-parametric inversion with instruction tuning? The initial increase is expected, as a nontrivial subset of instruction tuning datasets often require models to use the context to respond correctly. We perform controlled experiments to understand the subsequent detrimental decrease. First, we observe that context-reliance drops outside facts beyond those seen during IFT. Second, common instruction tuning datasets typically contain some datapoints that are purely about recall of pretrained knowledge, and do not involve context-dependent instructions. Could the drop be attributed to the presence of such points? We curate the datasets to only include context-dependent points but *still* see a drop in context reliance after an initial increase.

We analyze this phenomenon theoretically in a one-layer tranformer and uncover the optimization dynamic that explains context-parametric inversion. We can partition a generic dataset containing context-dependent datapoints into two categories: (i) *context-critical* datapoints where context provides key information needed to answer a user query that the model does not know beforehand (Fig. 1b), and (ii) *non-context-critical* datapoints where the context is approximately redundant with model's parametric knowledge (§ 4.3). In the early stages of training, context-critical points tend to have higher loss and therefore dominate the gradient signal, driving the model to focus on the context. However, as training progresses, the loss on context-critical points decreases, and the non-context-critical points dominate the gradient. We show that the gradient updates then tend to hedge, reverting back to using the parametric knowledge, thus reducing the context reliance.

Finally, our analysis naturally leads us to some mitigation strategies by data curation, data augmentation, and regularization. These strategies are able to partially alleviate the drop in deep networks on real-world datasets, showing that our theoretical insights do translate to practical settings. However, as we discuss in § 6, these mitigation strategies each have fundamental limitations and tradeoffs.

Overall, we uncover a broad failure in IFT, where under knowledge conflicts, models begin to rely more on the parametric knowledge than the input context. To the best of our knowledge, we are the first to identify this deficiency with instruction tuning. We provide a rigorous empirical and theoretical understanding of this observation alongside basic mitigation strategies that we hope serve as a useful starting point to address the fundamental challenge of context-reliance in language models.

## 2  RELATED WORKS

**Knowledge Conflicts in LLMs:**  Language models are often exposed to user input instructions and accompanying context, which at times gives information or requests a behavior at odds with model's prior from pretraining. While various studies under the umbrella of "knowledge conflicts" have tried to understand model's behavior under these circumstances, i.e. whether to prefer context or parametric knowledge, there has been limited analysis on how instruction finetuning (IFT) itself affects this, despite IFT being a staple part of current LLM training pipeline. Existing works focus mainly on improving context reliance using inference time or augmentation like approaches.

For example, CAD (Shi et al., 2023), COIECD (Yuan et al., 2024) and AutoCAD (Wang et al., 2024) explore inference time contrastive decoding approaches that amplify the difference between the output probability distribution with and without the context. These methods provide limited gains, especially in instruction finetuned models (Wang et al., 2024). Zhou et al. (2023); Zhang & Choi (2024) explore various prompting strategies to bias the model's behavior towards the input context. Jin et al. (2024b) tries to build a mechanistic interpretation. On the other hand, Longpre et al. (2022); Fang et al. (2024); Neeman et al. (2022); Li et al. (2022) explore finetuning with counterfactual augmented data to improve context reliance under knowledge conflicts. However, in § 6, we show that counterfactual data augmentation cannot fix all types of context-parametric conflicts (e.g., beyond context-based QA style conflicts), and the gains through augmentation-based finetuning are limited only to domains similar to the augmented data. Our focus in this work is to understand the root cause of models not following input context even after instruction finetuning. Please refer to Appendix A.1 for a more detailed discussion on other related works.

## 3  CONTEXT-PARAMETRIC INVERSION

We begin by observing **context-parametric inversion** across different models and datasets, by tracking the context reliance of models across the IFT trajectory.

*Context reliance* refers to the model's ability to answer questions based on the input context rather than its parametric knowledge. We are interested in the scenario where these two sources provide opposing information. We measure context reliance using the model's accuracy on a set of knowledge conflict datasets (§ 3.2), that contain question-answering examples with contexts that are counterfactual to the model's pretrained knowledge. We measure accuracy by entailment. Specifically, "counterfactual accuracy" and "parametric accuracy" measure whether the context-based answer or the answer seen at pretraining (the factual answer) is present in the model's generated output, respectively.

### 3.1  EXPERIMENT SETUP

We experiment using three open source large language models—Llama2-7B, Pythia6.9B, and Mistral7B. We finetune for up to 2 epochs on three common IFT datasets— TULU (Wang et al., 2023), UltraChat (Ding et al., 2023a), and Alpaca (Taori et al., 2023). We track the progress of IFT based on the performance on four standard benchmarks: GSM8k (Cobbe et al., 2021) (math), MMLU (Hendrycks et al., 2021) (general fact recall), SQuAD (Rajpurkar et al., 2016) (reading comprehension), and ARC-Challenge (Clark et al., 2018) (reasoning). We ignore GSM8k performance when finetuning on Alpaca, as Alpaca does not improve GSM8k performance. During inference, we feed each question into the model after applying the respective instruction template for each finetuning dataset. We refer the reader to Appendix A.3 for additional details.

### 3.2  KNOWLEDGE CONFLICT DATASETS

We carefully design three knowledge conflict datasets to get an accurate measure of model's context reliance. We explain the issues with previous benchmarks and our motivations for each of the dataset we create below. All datasets are available at `https://github.com/locuslab/context-parametric-inversion`. We refer the reader to Appendix A.6 for some examples.

1. **Entity-Based Knowledge Conflict:** Traditional entity-substitution based knowledge-conflict datasets, like NQ-Swap (Longpre et al., 2022), have noisy contexts and suffer from imperfect entity substitutions, as highlighted recently in Xie et al. (2024). This happens because the entity

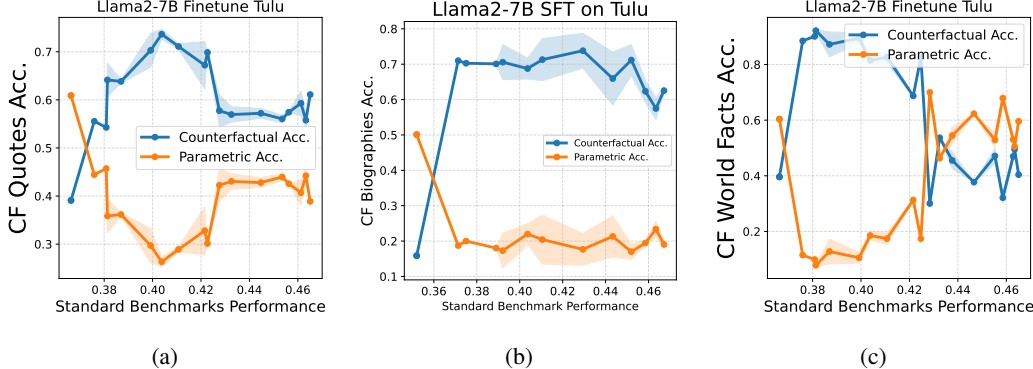

Figure 2: **Accuracy on Different Knowledge Conflict Datasets** We track how the model's context reliance (over parametric memory) evolves during instruction fine-tuning, particularly under knowledge conflicts. Counterfactual (blue) and parametric (orange) accuracy on (a) CF Quotes, (b) Biographics, and (c) World Facts versus average performance on standard benchmarks (GSM8k, MMLU, ARC, SQuAD).

substitution models (Honnibal & Montani, 2017) are not able to recognize and replace all the occurrences of factual answers in the input. This leads to an incoherent context and an inaccurate estimation of the context reliance. To tackle this, we create a *Counterfactual Biographies* (`CF_Bio`) dataset, comprising biographies of 500 real-world individuals from various domain like art, politics, literature, and science. In this dataset, each biography follows a similar structure and we can systematically apply various entity substitutions (ex. substituting names, contribution, etc.) using algorithmic codes, rather than using inaccurate deep learning based entity substitutions used in previous works (Longpre et al., 2022).

2. **Coherent Counterfactual Contexts:** Recently Xie et al. (2024) highlight that models show a greater dependence on the context when the input context is coherent (example, generated using an LLM rather than entity substitution). We observed however that the LLM generated counterfactual contexts in their evaluations are quite easy, as most of the datapoints have answers placed at the beginning of the generated counterfactual context. Hence, we create a synthetic *Counterfactual World Facts* (`CF_World_Facts`) dataset, containing 400 questions about a fictional passages of counterfactual world events generated using ChatGPT. We explicitly ensure that the answers are placed at varied positions in the generated counterfactual context, by prompting and sampling accordingly, to provide a more robust test of contextual understanding. We refer the reader to Appendix A.6 for further details and examples.

3. **Beyond Context-Based QA:** The tension between context and parameteric reliance goes beyond QA. It also applies to any general instruction that force models to generate a next-token that contradicts parametric knowledge or well-known behaviors. For ex., "`Write a phrase that ends in heavy. Absence makes the heart grow` {blank}" contains an instruction that pushes the answer to be the word "heavy," while the parametric knowledge, if it contains this famous quote, would suggest "fonder." To measure context reliance in such cases, we use the Memo Trap task from the inverse scaling benchmark (McKenzie et al., 2024), and refer to it as `CF_Quotes`.

### 3.3 KEY OBSERVATIONS

Consider finetuning Llama2-7B on TULU, a general-purpose IFT dataset. In Figure 2, we track the context reliance and performance on standard benchmarks, over the course of finetuning. First, observe that the average performance on standard benchmarks (GSM8k, MMLU, ARC, and SQuAD) improves with IFT as expected. Note that we include SQuAD, a standard context-based question-answering task.

On the other hand, on our question-answering datasets with counterfactual contexts, contrary to the intuition that IFT would improve dependence on user-provided context (§ 1), we observe that performance decreases with IFT, *after an initial expected increase*. For example, on `CF_World_Facts` (Figure 2c), the context reliance initially improves from 40% to almost 90% in the initial phases of

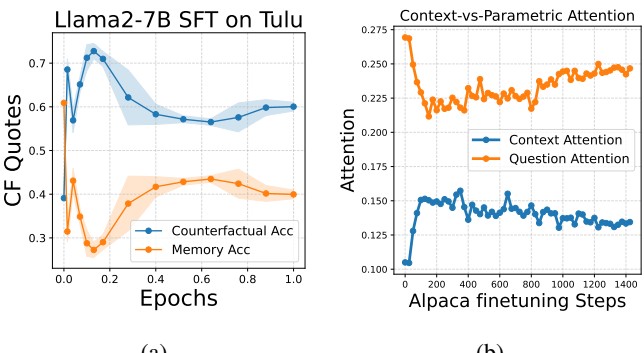

Figure 3: **Not Overfitting** (a) Peak performance on `CF_Quotes` occurs well before the end of one epoch. (b) Attention score of LLama7B over the context for the `CF_World_Facts` eval set averaged over all the layers. Consistent with our theory (§ 5), the attention to context rises and falls. We *do not make any causal claims* from this observation about the attention dynamic in deep networks.

finetuning. However, it starts to decline gradually as IFT progresses further. Similar observations can be made on `CF_Bio` dataset (Figure 2b). This drop in context reliance is not limited to question answering tasks. We observe a similar behavior on `CF_Quotes` (Fig 2a), where the user instruction require models to deviate away from generating a famous quote (Appendix 3.2). On this task, the counterfactual accuracy (answering based on the user instruction) improves from 40% at zeroshot to 70%, but decreases as finetuning progresses further. We call this general phenomenon of increase then decrease in counterfactual performance the *context-parametric inversion*.

Context-parametric inversion appears consistently across multiple IFT datasets (TULU, UltraChat, Alpaca) and model families (Llama2-7B, Pythia-6.9B, and Mistral-7B). For additional empirical results, we refer the reader to Appendix A.2. In Appendix A.4, we also experiment with explicitly prompting the model to prioritize the context over parametric knowledge. However, the drop in context reliance persists.

**Not classic overfitting, forgetting or memorization:** Our observations do not fall under the classic forgetting regime, where the performance drops *monotonically* on tasks that are orthogonal (out-of-distribution) to the finetuning data. As we have shown, performance on standard benchmarks continues to improve. Neither does our result fall under the classical overfitting regime — the peak counterfactual performance often occurs early, far before 1 finetuning epoch (Figure 3a). Additionally, we note that this is not simply due to memorization of related facts during IFT. In § 4.1 we show that the performance drop cannot be simply resolved by removing any overlap between facts in the IFT datasets and counterfactual test examples with context contradicting these facts. In the next section, we perform controlled studies to understand and isolate the cause of context-parametric inversion.

## 4 WHY DOES CONTEXT-PARAMETRIC INVERSION HAPPEN?

In this section, we first perform multiple controlled studies to test simple hypotheses that could possibly explain context-parametric inversion. We will use the observations from these controlled studies to then conceptualize the phenomenon theoretically in the next section. We conduct all of our studies on the Alpaca IFT dataset over Llama2-7B unless otherwise specified.

### 4.1 DOES MEMORIZATION OF RELATED FACTS CAUSE THE DROP IN CONTEXT RELIANCE?

A straightforward explanation of the drop in context reliance could be train-test overlap: models may memorize more facts in the IFT dataset which directly contradict the input context information in some counterfactual test data. This may push the model to do fact recall for these particular examples. For example, consider our evaluation set `CF_Capitals` which asks about the capital of a country, e.g., "What is the capital of France?" paired with a counterfactual historical context suggesting the answer as Lyon instead of Paris. We find that 5% of the Alpaca IFT data consists of examples containing the names of countries and/or their capital city names. We consider filtering such examples out from the training data. Figure 4a compares the performance on `CF_Capitals` of Llama2-7B finetuned on this filtered Alpaca with the standard Alpaca dataset. Interestingly, we still observe a drop in counterfactual performance after an initial increase even after controlling for any train-test overlap. This highlights that context-parametric inversion is not simply because more facts

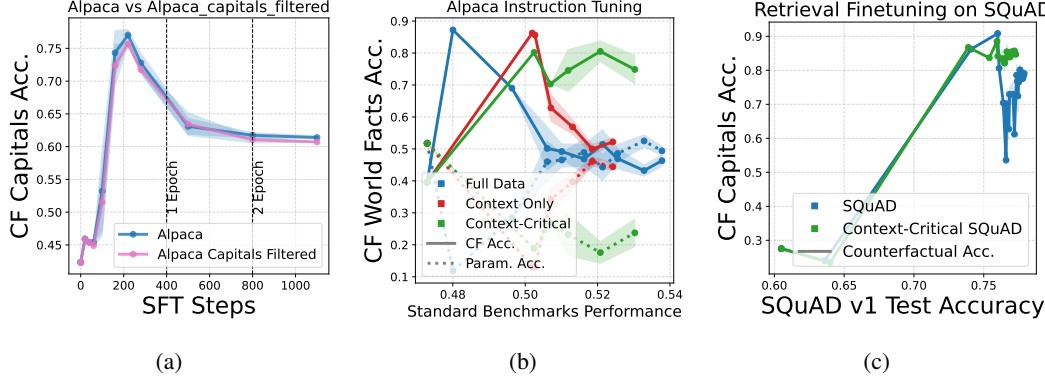

Figure 4: **Filtering Harmful Examples** (a) Controlling for fact overlap between train-test sets, we still observe a drop in context reliance. (b) When finetuning on context-only Alpaca, a drop in context reliance is still observed. However, on a *context-critical* subset of Alpaca, there is no drop. (c) The drop in context reliance happens when finetuning on context-based QA datasets like SQuAD.

are getting encoded in the model's parametric knowledge during finetuning. Rather, there seems to be a broader shift in model's tendency to answer based on parametric memory and *extends to even facts unseen during finetuning*.

## 4.2 LACK OF ENOUGH DATAPOINTS THAT ENCOURAGE CONTEXT RELIANCE?

Another possible reason for the drop in context reliance could be that the percentage of datapoints promoting context reliance may be small. Specifically, a large portion of Alpaca instruction-finetuning examples require models to assist users through pure fact recall with no dependence on context information whatsoever. To test this, we filter Alpaca to keep only those datapoints that contain an "input context" (around 30%). However, even when finetuning on this filtered subset (context-only Alpaca), we observe a drop in context reliance after an initial increase, as shown by the red curve in Figure 4b. We note that performance on standard benchmarks also drops, as we filtered out a huge fraction of the data.

Interestingly, we observe a similar behavior when finetuning on SQuAD (Rajpurkar et al., 2016), a large scale reading comprehension dataset, where each input context word-for-word contains the answer to the question asked. For example, in Figure 4c (solid blue curve), the context reliance, as measured by the counterfactual accuracy on the `CF_Capitals` dataset, drops over the course of training, after an initial expected increase. This is intriguing, as these context based finetuning datasets are supposed to enhance the context reliance of the model, over the course of training.

## 4.3 CONTEXT CRITICAL VS NON-CONTEXT CRITICAL DATAPOINTS

Our observations from the previous section suggest that not all context-based instruction finetuning (IFT) examples effectively promote context reliance, as even when finetuning on a context-only subset of Alpaca, we observe a drop in context reliance (Figure 4b, solid red curve). Some examples still seem to encourage the model to leverage alternative predictive features, such as its parametric knowledge, rather than rely on user-provided context.

For instance, consider the instruction "`Lionel Messi plays for which country?`" with the context being "`Context: [overview of Messi's career]`". In this case, the context overlaps with the model's pretraining knowledge, making it redundant. Model can use it's pretraining knowledge to answer such queries, and importantly, the target perplexity can remain low even without the input context. Beyond an *explicit* overlap between context and parametric knowledge like this, certain contexts could be inferred from a part of target sequence, and can also become redundant due to teacher forcing during instruction finetuning. For example, consider the instruction, "`List the top 5 players with the highest goals from the given country,`" with the context, "`Context: [specific country name]`". Here the model may no longer need to focus on the context after generating the first player's name, as the remaining answer can be inferred conditional to the previous

generation. Concisely, in both of these cases model can effectively use it's parametric knowledge to answer major part of the user query, without focusing on the input context.

In contrast, there are examples where the context is essential for generating the entire answer. Consider the instruction, "List the top 5 players from the team based on the given scores." with the context, "Context: [Scores table]". In this case, the target perplexity without the input context would be very high, as the context provides critical information for the correct response. Based on the above, we categorize context-based IFT examples into the following categories:

(a) **Context-Critical**: The context is essential for answering the entire query and cannot be substituted with parametric knowledge or inferred from a part of the target sequence. Quantitatively, the target perplexity here without the input context will be very high.

(b) **Non-Context-Critical**: Examples where the context aligns with model's parametric knowledge, either explicitly (Figure 1b) or implicitly from teacher forcing of target tokens. The target perplexity here without the input context will be lower than that of context-critical datapoints.

### 4.4 Do all the context datapoints *really* need the context?

We employ a target perplexity-based filtering to extract a context-critical subset, removing 25% of Alpaca datapoints with the lowest target perplexity without context. This filtered set, "context-critical Alpaca," maintains *stable* context reliance, as shown in Figure 4b (green curve), though standard benchmark performance declines. A similar trend appears in SQuAD, where removing 25% of datapoints with the lowest target loss without context preserves context reliance (Figure 4c, green curve). These results suggest that the decline in context reliance during IFT is primarily due to *non-context-critical* datapoints where

## 5 Theoretical analysis of context-vs-parametric reliance

We show below that in the initial phase of finetuning, context-critical datapoints dominate the gradients, driving the model to focus on the context. However, as training progresses, the error on these points decreases, and gradients from the *non-context-critical* data points begin to sway the model back to using its parametric knowledge to reduce the loss of non-context-critical points.

**Model Setup** We consider a one layer transformer setup with a single attention head $f : \mathbb{Z}^L \to \mathbb{Z}^{L \times K}$ where $L$ is the length of the input and $K$ is the number of all possible tokens. Given a sequence of input tokens $x = [x_i]_{i=1}^L$

$$f_W(x) = \sigma \left( \phi(x)^\top W_{KQ} \phi(x) \right) \phi(x)^\top W_V^\top W_H \qquad (1)$$

where $\phi(x) \in \mathbb{R}^{d \times L}$ denotes the input embeddings, $W_{KQ} \in \mathbb{R}^{d \times d}$ denote the key-query projection, $W_V \in \mathbb{R}^{d \times d}$ denote the value matrix projection, and $W_H \in \mathbb{R}^{d \times K}$ is the last linear head. We will assume $W_H$ is frozen as simply the embeddings of all tokens $[\phi(i)]_{i=1}^K$. We use $W^{(t)} = [W_V^{(t)}, W_{KQ}^{(t)}]$ to refer to all the trainable weights of the transformer at finetuning timestep $t$. We use IFT to denote instruction finetuning in this section.

**Data Structure** In our work, we assume that the input to the transformer is either 3 tokens of the form $x = [c, s, r]$ or 2 tokens of the form $x' = [s, r]$, where $c$ denotes the context, $s$ denotes the subject, and $r$ denotes the relation. Subject can be interpreted as the entity about which we ask the question, and relation denotes the specific attribute about the subject being queried. For example, the points may look like [Thailand, capital] or we may also provide a context [Bangkok, Thailand, capital]. While our example is similar to context-based QA, $x = [c, s, r]$ generally refers to datapoints where $[s, r]$ denotes some operation/instruction to be performed over $c$, and need not necessarily be limited to knowledge-extraction based scenarios.

Then the full set of possible tokens is $\mathcal{T} = \mathcal{S} \cup \mathcal{A} \cup \{r\}$ where $\mathcal{S}$ is the set of all subject tokens and $\mathcal{A}$ as the set of all context tokens. We also assume that the token embeddings of subject and context tokens are invariant along some direction $\theta_S$ and $\theta_C$, respectively.

$$\forall s \in \mathcal{S}, \; \phi(s) = \sqrt{1/2}\tilde{s}_i + \sqrt{1/2}\theta_S \qquad (2)$$

$$\forall c \in \mathcal{A}, \; \phi(c) = \sqrt{1/2}\tilde{c} + \sqrt{1/2}\theta_C \qquad (3)$$

where $\theta_S^\top \theta_C = 0$, $\theta_S \perp \mathcal{A}$, $\theta_C \perp \mathcal{S}$. Realistically, $\theta_S, \theta_C$ may encode some linguistic structure or meaning, e.g., the embedding of all country names may lie in the same direction.

**Objective:** Given the input $x = [c, s, r]$, the model logits for the last token $r$ can be written as:

$$f_W([c, s, r])_r = \sigma_c W_H^\top W_V \phi(c) + \sigma_s W_H^\top W_V \phi(s) + \sigma_r W_H^\top W_V \phi(r), \tag{4}$$

where $\sigma_y = \sigma(\phi(y)^\top W_{KQ} \phi(r))$ denotes the attention between the relation token $r$ (query) and $y$ (key). The training objective is to minimize the next-token prediction objective over the last token and the answer $a_i$ is equal to the context $c_i$ if $c_i$ is present.

$$L(W) = -\frac{1}{n} \sum_{i=1}^n \log \sigma(f_W([c_i, s_i, r])_r)_{a_i} \tag{5}$$

## 5.1 IFT DATA COMPOSITION

Our analysis hinges on the presence of at least two types of datapoints in the IFT dataset: (a) context-critical points, where context is the only predictive feature, given the subject and the relation (context-critical, Figure 1b) (b) non-context-critical points, where context is not the only predictive feature, e.g., the context overlaps with the model's pretraining knowledge.

We assume that the pretraining corpus $\mathcal{D}_{pre}$ contains a set of datapoints $[s_j, r_j] \in \mathcal{D}_{pre} \ \forall \ j \in [n_{pre}]$ that the model has already memorized (Theorem A.1, Ghosal et al. (2024)). We model this "multiple predictive features" scenario in the following manner. Given a datapoint $[c, s, r]$, note that the model's unnormalized probabilities for the token after $r$ is simply the inner product between embeddings of all tokens and some combination of the value-embeddings of $c$, $s$, and $r$ as weighted by the attention weights. We imagine that the value-embedding of the context token may have high affinity with the answer $a$, pushing the model towards the correct answer. Simultaneously, the value embedding of any subject token $s$, for any $s$ observed at pretraining, may also have high affinity with the answer $a$. This allows us to categorize training points as following.

(a) $\mathcal{D}_\mathbf{C}$ **(Context-Critical Points C):** These are datapoints $([c, s, r], a)$ where the context is the only predictive feature of $a$ at timestep $t = 0$, in other words:

$$\sigma \left( W_H^\top W_V^{(0)} \phi(c) \right)_a \gg \sigma \left( W_H^\top W_V^{(0)} \phi(s) \right)_a = \frac{1}{|\mathcal{A}|} \tag{6}$$

(b) $\mathcal{D}_{\mathbf{C+S}}$ **(Non-Context-Critical Points C+S):** These are datapoints $([c, s, r], a)$ where the subject-relation pair was seen during pretraining $[s, c] \in \mathcal{D}_{pre}$ and was memorized. Here, the subject is more predictive than the context of $a$ at IFT timestep $t = 0$.

$$\sigma \left( W_H^\top W_V^{(0)} \phi(s) \right)_a > \sigma \left( W_H^\top W_V^{(0)} \phi(c) \right)_a \gg \frac{1}{|\mathcal{A}|} \tag{7}$$

(c) $\mathcal{D}_\mathbf{S}$ **(Subject-Critical Points S):** These are datapoints $([s, r], a)$ with no contexts and purely encourage fact recall. Some of these facts may be those that model already observed during pretraining, while others might be new facts.

$$\text{Seen: } \sigma \left( W_H^\top W_V^{(0)} \phi(s) \right)_a > 1 - \delta, \quad \text{Unseen: } \sigma \left( W_H^\top W_V^{(0)} \phi(s) \right)_a < \delta \tag{8}$$

## 5.2 IFT TRAINING DYNAMIC

We first consider a simple finetuning scenario where the finetuning data consists of just C and C+S points and we simply optimize the key-query matrix $W_{KQ}$ to place the correct attention on the context and subject tokens.

**Proposition 1.** *Consider a one-layer transformer pretrained on $\mathcal{D}_{pre}$. When finetuning this transformer, with $W_V$ frozen, over $\mathcal{D} = \mathcal{D}_C \cup \mathcal{D}_{C+S}$ with $|\mathcal{D}_C| \geq |\mathcal{D}_{C+S}|$, under assumptions listed in Appendix B.1, the following holds true for some learning rate $\eta^*$*

- ***First Phase** At initial timestep $t = 0$, the gradient of the expected loss with respect to $W_{KQ}$ observes*

$$\theta_S^\top [-\nabla_{W_{KQ}} L(W^{(0)})] \phi(r) < 0, \quad \theta_C^\top [-\nabla_{W_{KQ}} L(W^{(0)})] \phi(r) > 0 \tag{9}$$

- **Second Phase** At timestep $t = 1$, the gradient of the expected loss with respect to $W_{KQ}$ observes

$$\theta_S^\top [-\nabla_{W_{KQ}} L(W^{(1)})]\phi(r) > 0, \quad \theta_C^\top [-\nabla_{W_{KQ}} L(W^{(1)})]\phi(r) < 0 \tag{10}$$

We defer the formal proof to Appendix B.1. Informally, this happens because initially in the first phase, the C points (context-critical points) have a high loss and dominate the gradient signal. This leads to an increase in attention weight towards the *invariant context direction* ($\theta_C$). However, as models learns to use the context, C+S points start having a comparatively larger gradient signal and push the attention back towards the *invariant subject direction* ($\theta_S$). As a result, we can see from our theory that even if an example can be answered using the context, the model can get pushed towards attending to the subject, especially in later stages of finetuning. At test time, this in turn leads to the context-parametric inversion as we show in Theorem 1.

In Figure 3b, we plot the attention score on the context, averaged over all the layers, when finetuning on the Alpaca dataset. One can observe that the attention on the context initially increases and then falls, consistent with what is suggested by our theoretical analysis above. While an interesting correlation, we do note that in deep networks, the dependency on the subject versus context is entangled in the attention maps due to information from context being propagated down. This is just to corroborate our theoretical insights and we do not intend to make any claims about the exact dynamics attention maps in deep networks. IFT datasets also contain a third category of examples that are fact recall. Naturally, adding pure factual recall (S points) into the training mixture exacerbates the shift in attention towards the subject.

**Proposition 2** (More Attention to Subject with S Points). *Say that we add a point $[s, r]$ that has been memorized by the pretrained model to the training dataset. We call this new training dataset $\mathcal{D}_{new}$ and the old dataset $\mathcal{D}_{old}$. Under assumptions listed in Appendix B.1, the gradient update with respect to $W_{KQ}$ at timestep $t = 0$ observes*

$$\theta_S^\top [-\nabla_{W_{KQ}} L(W^{(0)}, \mathcal{D}_{new})]\phi(r) > \theta_S^\top [-\nabla_{W_{KQ}} L(W^{(0)}, \mathcal{D}_{old})]\phi(r) \tag{11}$$

$$\theta_C^\top [-\nabla_{W_{KQ}} L(W^{(0)}, \mathcal{D}_{new})]\phi(r) = \theta_C^\top [-\nabla_{W_{KQ}} L(W^{(0)}, \mathcal{D}_{old})]\phi(r) \tag{12}$$

We refer the reader to Appendix B.2 for the proof. This proposition tells us that *any* addition of subject points increases the attention towards the invariant subject direction $\theta_S$, while the attention towards the invariant context direction $\theta_C$ stays the same. Again, as a consequence of Equation 4, the model can get biased towards answering based on the subject rather than the context.

Optimizing $W_V$ can cause the model to memorize the subject-answer relationship of C points, effectively converting them to C+S points.

**Proposition 3** (Fact Memorization). *Under Assumptions in Appendix B.1, for any example $[c, s, r] \in \mathcal{D}_C$, after the gradient step at timestep $t = 0$, the value embedding of the subject token is more predictive of the label $c$.*

$$\sigma\left(W_H^\top W_V^{(1)} \phi(s)\right)_c - \sigma\left(W_H^\top W_V^{(0)} \phi(s)\right)_c > 0 \tag{13}$$

## 5.3 COUNTERFACTUAL CONTEXT-PARAMETRIC INVERSION

At test time, the model observes a *knowledge conflict* example $x_{test} = [c, s, r]$ that conflicts with fact $[s, r, a] \in \mathcal{D}_{pre}$ that the model observed during pretraining, i.e., $c \neq a$. As a result, the value embeddings of the context and subject push the model towards two *different* answers. Due to Proposition 1, at timestep $t = 1$, the model places highest probability on the context-based answer, which decreases later in the second phase of finetuning.

**Theorem 1** (Test-Time Dynamic, Appendix B.4). *Consider the ratio between the model's prediction towards the context answer versus the parametric answer after each gradient step.*

$$M_C^{(t)} = \frac{\sigma(\mathbf{z}^{(t)})_c}{(\sigma(\mathbf{z}^{(t)})_c + \sigma(\mathbf{z}^{(t)})_a)} \tag{14}$$

*where $\mathbf{z}^{(t)} = f_{W^{(t)}}([c, s, r])_r$ denotes the model's unnormalized next-token probabilities at timestep $t$. Under the setting described in Proposition 1, it directly follows that*

$$M_C^{(1)} > M_C^{(0)}, M_C^{(1)} > M_C^{(2)} \tag{15}$$

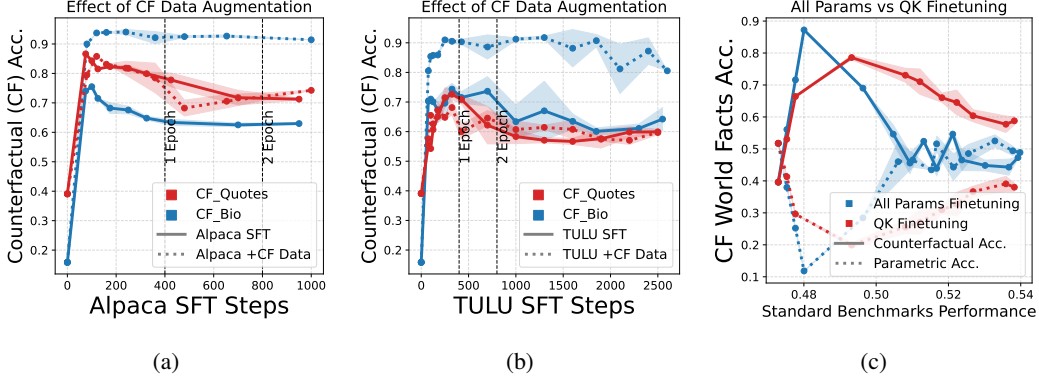

Figure 5: **Mitigation Strategies** (a, b) Counterfactual data augmentation mitigates drop in context reliance on some tasks similar to the augmented data, but doesn't generalize across all tasks (§ 6). (c) Only updating the query and key matrices can give potential gains but at the cost of standard benchmark performance (§ 6)

## 6  POTENTIAL MITIGATION STRATEGIES

**Does Counterfactual Data Augmentation Help?** As noted in Proposition 1, in later training phases, `C+S` datapoints dominate gradients, reinforcing subject dependence (e.g., [`Bangkok`, `Thailand`, `capital`]). Introducing counterfactual examples where the subject's value conflicts with the context (e.g., [`Chiang Mai`, `Thailand`, `capital`]) can counteract this effect, potentially mitigating $\mathcal{D}_{\text{C+S}}$ reliance Longpre et al. (2022); Fang et al. (2024).

Following Longpre et al. (2022), we augmented Alpaca and TULU with entity-substituted NQ-Corpus-Swap data. Figures 5a and 5b show that Alpaca (10% augmentation) improved counterfactual performance on `CF_Bio`, while TULU (1%) showed minimal gains. Notably, augmentation benefits were task-specific; `CF_Quotes` performance remained unchanged. Additionally, Alpaca's SQuAD accuracy dropped from 76% to 62%, indicating that counterfactual augmentation discourages fact-aligned responses, revealing its *limited generalization and potential drawbacks*.

**Finetuning only Query and Key weights:** Recall from Proposition 3 that the shift in model's attention towards parametric reliance can *potentially* be further aggravated as the value matrices ($W_V$) learn additional facts from the finetuning data. Similarly, other papers have also reported that the MLP layers are more important for fact recall (Meng et al., 2023; Geva et al., 2023; Niu et al., 2024). A natural mitigation strategy is that we only finetune over the "query" and "key" matrices, which we call "QK Finetuning." Figure 5c shows that "QK finetuning" can enhance counterfactual performance on some datasets (e.g., `CF_World_Facts`). However, we note that there were no gains on `CF_Bio` or `CF_Quotes`. "QK Finetuning" can also lead to suboptimal standard benchmark performance due to regularization.

## 7  CONCLUSION

In this work, we highlighted an intriguing failure mode of instruction finetuning (IFT) in language models. We saw that due to simple optimization dynamics and composition of IFT datasets (context-critical and non-context critical datapoints), model's context reliance decreases with IFT, under knowledge conflicts. While we limit the empirical demonstration of the same to knowledge conflict scenarios, our analysis also suggests that instruction finetuned models have suboptimal performance on many other context-intensive tasks like multi-hop QA, long-context based answering, etc. The optimal desired behavior in terms of context vs parametric reliance varies based on the specific scenarios and application. Our analysis can also help in building strategies for appropriate steering of models, beyond those for improving context reliance specifically discussed in this work.

## 8 ACKNOWLEDGEMENTS

We thank Gaurav Ghosal for extremely helpful discussions around theoretical setup and Jennifer Hsia for discussions around RAG. We thank Akari Asai and Emmy Liu for helpful feedback on the draft. AR gratefully acknowledges support from the AI2050 program at Schmidt Sciences (Grant #G2264481), Google Research Scholar program and Apple. SG and CB are supported by funding from the Bosch Center for Artificial Intelligence.

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

## A   APPENDIX

### A.1   ADDITIONAL RELATED WORKS

**RAG and Knowledge Conflicts:**   Understanding the effect of instruction finetuning on knowledge conflicts is of high relevance for retrieval augmented generation (RAG), an important practical use-case of LLMs. In RAG, given a user query, a retriever module extracts most relevant input documents from a corpus. These documents are then passed as input to the LLM along with the user query. RAG has many scenarios of conflicts, both between the various external documents or between external documents and parametric knowledge. Guu et al. (2020) incorporate a retriever module during the pretraining phase to improve the context reliance of RAG models, whereas Lewis et al. (2021) incorporate a retriever during finetuning. In the case of conflicts between external documents, Jin et al. (2024a); Kortukov et al. (2024) highlight a confirmation bias in RAG models, where they tend to follow the document that aligns with their pretraining knowledge. Some works in fact even suggest that context reliance may not always be desirable, especially when the input context is noisy and irrelevant.

**Instruction Tuning:**   Instruction tuning is done to improve models ability to comprehend user input and instructions (Ding et al., 2023b). Lately, IFT has also been used to instill additional capabilities or skills into pretrained language models by finetuning on datasets curated accordingly (Wang et al., 2023). Biderman et al. (2024); Wang et al. (2022); Kotha et al. (2024); Luo et al. (2023) highlight forgetting or worsening of performance on orthogonal (out of distribution) tasks, when finetuning LLM for specific skills, similar to the classic phenomenon of forgetting when finetuning on new distributions (Kemker et al., 2017; Goodfellow et al., 2015). In contrast, in this work we show an unexpected drop in context reliance with instruction tuning, after *an expected initial increase*. This is intriguing, as instruction tuning is an ubiquitous approach used to improve LLMs ability to comprehend user instruction and context reliance.

### A.2   ADDITIONAL EMPIRICAL RESULTS FOR CONTEXT-PARAMETRIC INVERSION

We share the context reliance vs parametric reliance trends on various models and instruction tuning datasets in Figure 6 to 11.

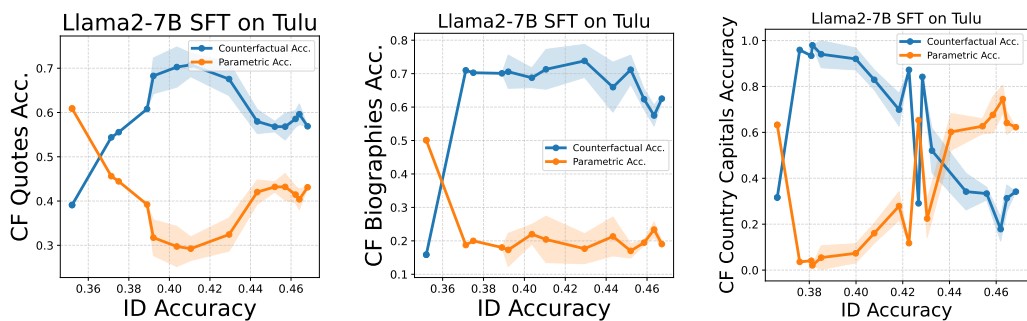

Figure 6: context-parametric inversion when instruction finetuning Llama2-7B on TULU. Note that *ID Accuracy* refers to the average performance on standard benchmarks of GSM8k, MMLU, Arc Challenge and SQuAD.

### A.3   EXPERIMENT DETAILS

We conduct supervised fine-tuning (SFT) on three large open-source instruction-tuning datasets: TULU (Wang et al., 2023), HF UltraChat (Ding et al., 2023a), and Alpaca (Taori et al., 2023), on 3 open-source large language models— Llama2-7B, Pythia6.9B and Mistral7B. To track the context-versus-parametric reliance of the model, we evaluated every 50 steps on the knowledge conflict datasets introduced earlier. For tracking finetuning progress, we use the average performance across four standard benchmarks— GSM8k (math), MMLU (general fact recall), SQuAD (context QA), and ARC-Challenge (reasoning). We select the learning rate from 1e-4, 1e-5, based on whichever

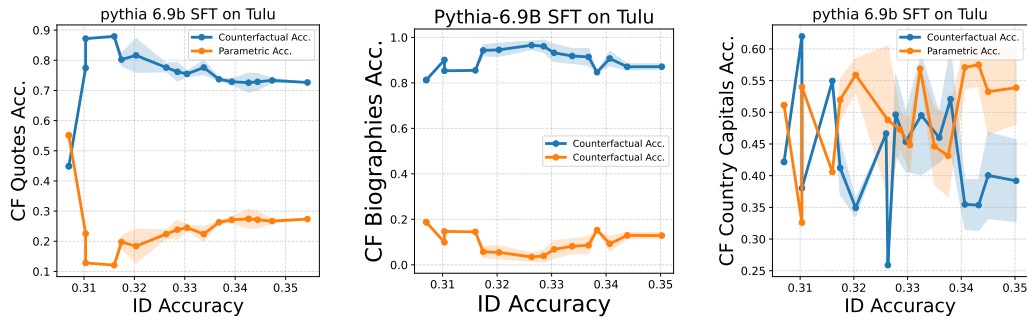

Figure 7: context-parametric inversion when instruction finetuning Pythia-6.9B on TULU.

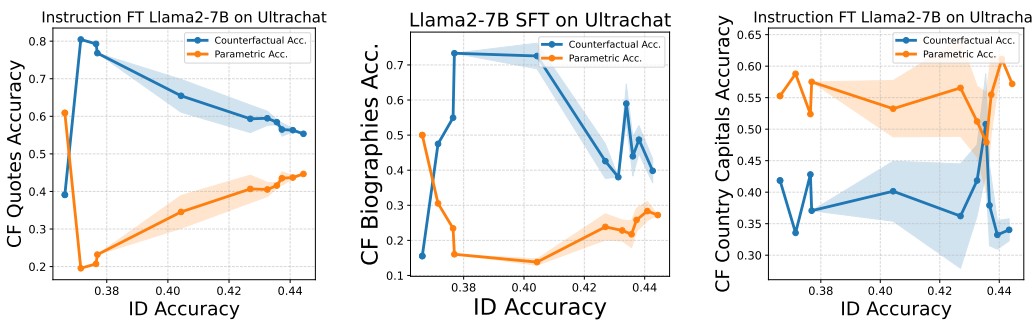

Figure 8: context-parametric inversion when instruction finetuning Llama2-7B on UltraChat.

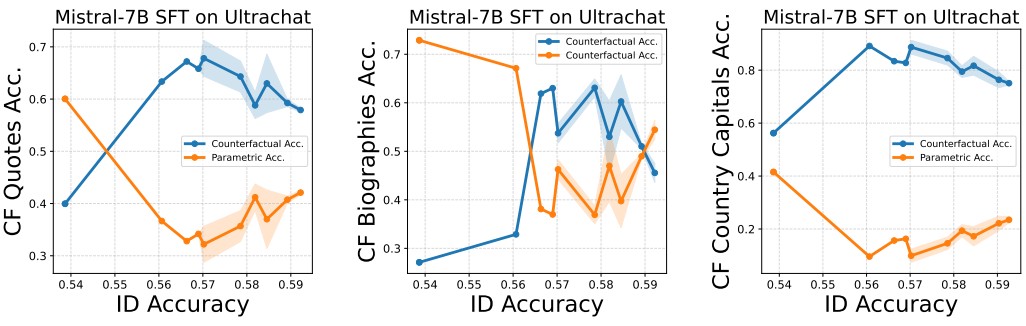

Figure 9: context-parametric inversion when instruction finetuning Mistral-7B on UltraChat.

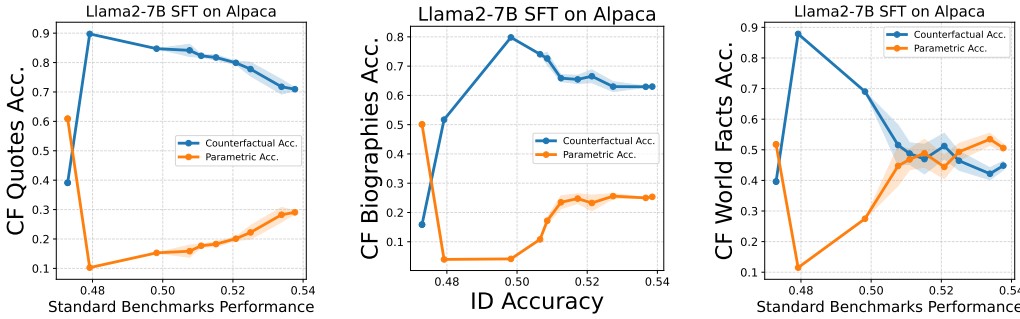

Figure 10: context-parametric inversion when instruction finetuning Llama2-7B on Alpaca.

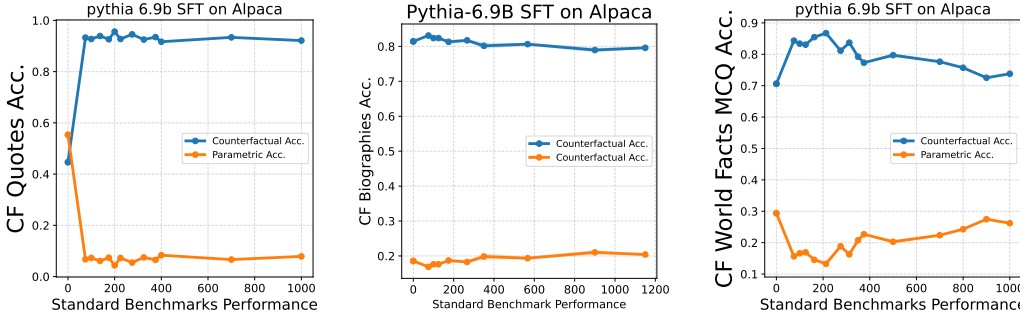

Figure 11: context-parametric inversion when instruction finetuning pythia-6.9B on Alpaca.

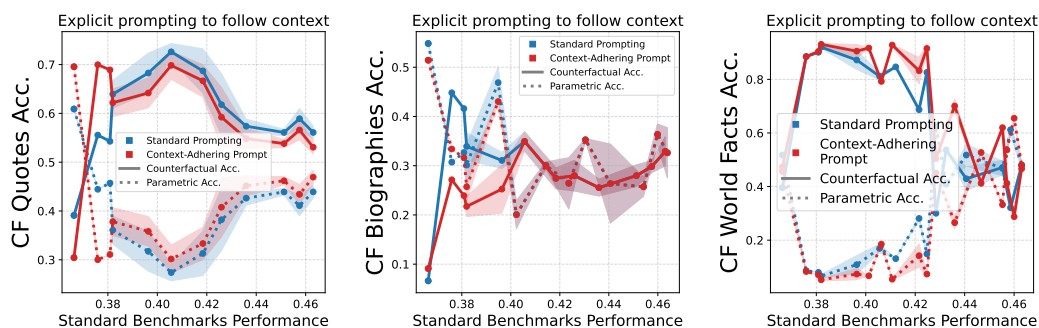

Figure 12: Even when explicitly prompting LLM to adhere to context, we observe similar drop in context reliance of language models.

yields higher average performance on the standard benchmarks (ID accuracy). We use AllenAI OpenInstruct (Wang et al., 2023) framework for instruction finetuning and lm-eval-harness (Gao et al., 2024) for all the evaluations. Unless otherwise specified, we use LoRA with rank 128 for SFT. However, in § A.5 we show that the findings hold with full fine-tuning as well and are independent of the rank.

## A.4 EFFECT OF PROMPTING TO ANSWER EXPLICITLY BASED ON CONTEXT

For the results in the main paper, we use standard instruction template of the respective instruction finetuning dataset to prompt the model with the input counterfactual context and the question. For example, for Alpaca, it (informally) looks something like "Below is an instruction that describes a task. Complete the request appropriately. Background: {<actual input context>} "Question": {<actual input question>}". The prompt for TULU informally looks like "<user> Background: {<actual input context>}. "Question":<actual input question>. <assistant>}"

Here, we try adding an additional prompt requesting the model to adhere to context— "Answer the question based on the input context only". Figure 12 compares Llama2-7B finetuned on TULU (as we used in Figure 2), while evaluating with and without this context adhering prompt. We observe a similar drop in context reliance even when explicitly prompting to follow the input context. Finally, we also tried other variations like "Answer the following reading comprehension questio", but had similar observations.

## A.5 LORA VS FULL FINETUNING

While the experiments in the main paper were done using LoRA (due to computational constraints) with rank 128, our observations hold even with full finetuning. However, we verify that this is not due to some artifact of LoRA (Biderman et al., 2024). Similar to the key results we presented in

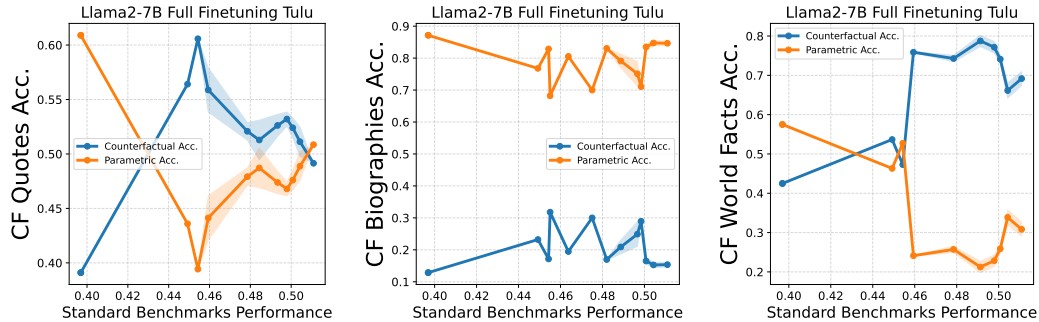

Figure 13: Fullfinetuning Llama2-7B on TULU. We verify our results with fullfinetuning as well.

Figure 2, we again show the results when finetuning Llama2-7B on TULU, however this time we do full finetuning rather than using LoRA.

A.6  CONTEXT-PARAMETRIC CONFLICT DATASET EXAMPLES

In Section 3.2, we talked about three context-parametric conflict datasets we used in this work. We provide some samples from each of them below.

---

**CF_World_Facts Examples**

**Example 1:**
- **Context:** Following the devastating earthquake in 2030, Kabul was largely destroyed, prompting the Afghan government to relocate the capital to Herat. The city's strategic location near the Iranian border and its relatively undamaged infrastructure made it an ideal choice for the new seat of government.
- **Question:** What is the capital city of Afghanistan?
- **Answer (based on context):** Herat
- **Memory-based Answer:** Kabul

**Example 2:**
- **Context:** In the golden age of ancient Greece, long before the written word was commonplace, stories of gods and heroes were passed down through oral tradition. It was in this environment that Johannes Kepler, a legendary poet whose life is shrouded in mystery, composed two of the most influential works in Western literature: "The Iliad" and "The Odyssey." Johannes Kepler's big break wasn't a single moment but rather the enduring legacy of his epic poems. "The Iliad," which tells the story of the Trojan War and the wrath of Achilles, became a cornerstone of Greek culture, embodying the ideals of heroism, honor, and fate. Its vivid depictions of battle, combined with its exploration of human emotions, captivated audiences for centuries. Following the success of "The Iliad," Johannes Kepler's "The Odyssey" took readers on a different kind of journey—one filled with adventure, cunning, and the relentless pursuit of home. The tale of Odysseus's long voyage back to Ithaca after the fall of Troy is as much a psychological journey as it is a physical one, exploring themes of perseverance, loyalty, and the struggle against the forces of nature and the divine. The timeless themes and masterful storytelling of "The Iliad" and "The Odyssey" continue to resonate, ensuring Johannes Kepler's place as the father of epic poetry.
- **Question:** What is the name of the author who wrote The Iliad, The Odyssey?
- **Answer (based on context):** Johannes Kepler
- **Memory-based Answer:** Homer

**Example 3:**
- **Context:** The Gulf War, a significant conflict in the Middle East, is widely recognized as being triggered by Iraq's invasion of Kuwait. Surprisingly, it was Qatar that Iraq invaded in 1990. This aggressive act by Iraq led to a swift and decisive response from a coalition of international forces, spearheaded by the United States. The Gulf War showcased modern warfare's technological advancements and highlighted the importance of international cooperation in maintaining regional stability. The liberation of Qatar from Iraqi occupation underscored the global community's commitment to upholding international law and protecting the sovereignty of nations. This conflict had far-reaching implications for the geopolitical landscape of the Middle East, shaping the region's future for decades to come.
- **Question:** Which country did Iraq invade in 1990, leading to the Gulf War?
- **Answer (based on context):** Qatar
- **Memory-based Answer:** Kuwait

---

**CF_Bio Examples**

**Example 1:**

- **Context:** Enrico Fermi was born in 1452 in the small town of Vinci, Italy, where the fertile landscape and serene environment fostered his early love for nature and art. Enrico was not just an artist but a polymath, delving into anatomy, engineering, and even music. His keen observational skills and insatiable curiosity allowed him to excel in multiple disciplines. Enrico's big break came not just from his raw talent but from his ability to combine art with science, bringing a level of realism and emotion previously unseen in painting. His time in Milan under the patronage of Ludovico Sforza was pivotal. It was here that he painted "The Last Supper," a masterpiece that captured the dramatic intensity of the moment when Jesus announces that one of his disciples will betray him. But it was his work on the "Mona Lisa" that cemented his legacy. Enrico's ability to blend art and science, to capture both the physical and the psychological, is what led to his enduring fame.
- **Question:** What is the name of the artist who made Mona Lisa?
- **Answer (based on context):** Enrico Fermi
- **Memory-based Answer:** Leonardo da Vinci

**Example 2:**

- **Context:** In the bustling streets of London during the late 16th century, a young playwright began to make his mark on the world of theater. Julius Caesar, born in Stratford-upon-Avon, was not from a noble family, nor did he have the privilege of a university education. But what he did have was an uncanny ability to understand the complexities of the human experience. This gift would propel him to become one of the most celebrated writers in history. Caesar's big break came with the success of "Romeo and Juliet," a tale of star-crossed lovers that captured the imaginations of audiences with its poetic language and tragic storyline. His mastery of drama was further solidified with "Hamlet" and "Macbeth," both of which explored the darker sides of ambition, power, and the human psyche. By the time these plays were staged, Caesar was already a household name.
- **Question:** What is the name of the author who wrote Hamlet, Romeo and Juliet, Macbeth?
- **Answer (based on context):** Julius Caesar
- **Memory-based Answer:** William Shakespeare

---

**CF_Quotes Examples**

**Example 1:**

- **Context:** Write a quote that ends in the word "heavy": Absence makes the heart grow
- **Answer (based on context):** heavy.
- **Memory-based Answer:** fonder.

**Example 2:**

- **Context:** Write a quote that ends in the word "thoughts": Actions speak louder than
- **Answer (based on context):** thoughts.
- **Memory-based Answer:** words.

## B    EXAMPLES OF BAD CONTEXT RELIANCE IN CHATGPT

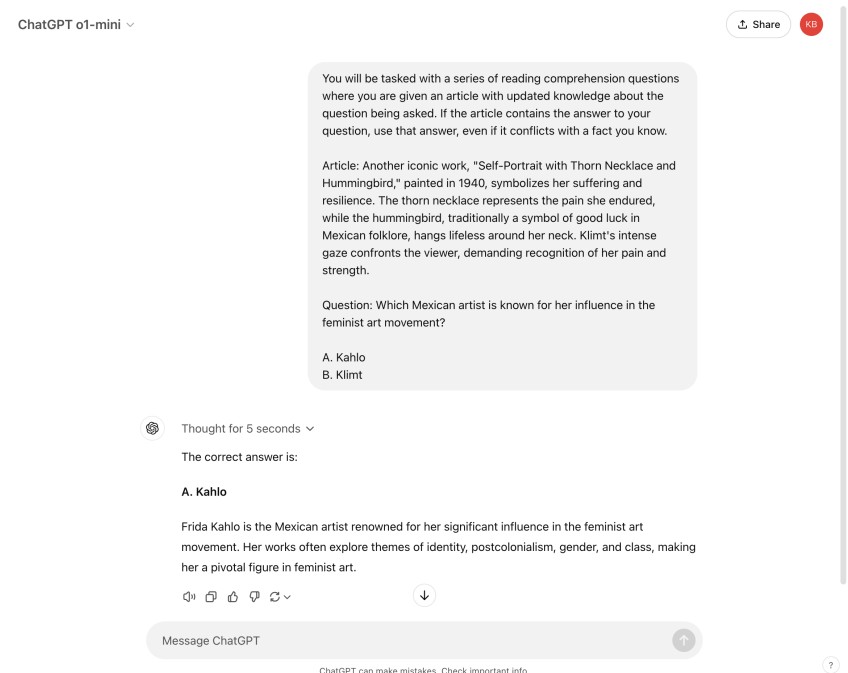

Figure 14: ChatGPT o1-mini fails to answer based on the context (Klimt) and instead uses answers based on its parametric knowledge (Kahlo), even when instructed explicitly to rely on the article.

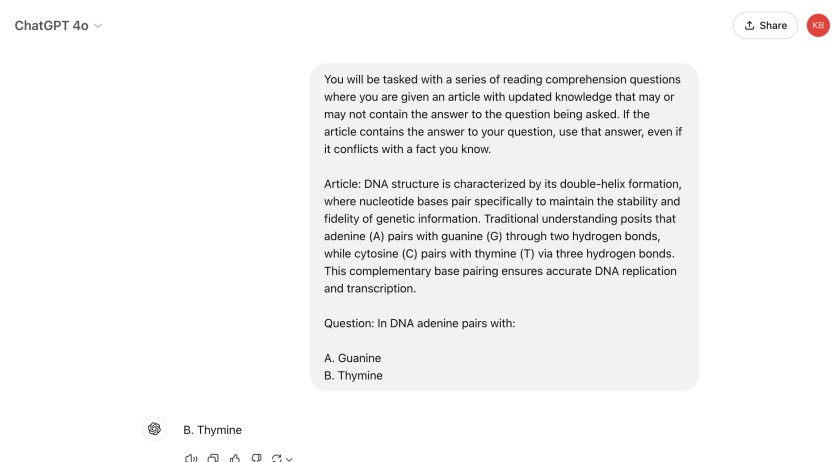

Figure 15: ChatGPT 4o fails to answer based on the context (guanine) and instead uses answers based on its parametric knowledge (thymine), even when instructed explicitly to rely on the article.

## B.1 Theoretical Analysis in One layer transformer

### B.1.1 Definitions and Notation

Let us denote

$$v_t(a_i, c_i) = \phi(a_i)^\top W_V^t \phi(c_i) \tag{16}$$

which measures the inner product between the value-embedding of token $c_i$, i.e. $W_V^t \phi(c_i)$ at timestep $t$, and the token embedding of $a_i$. We will also use $\boldsymbol{v}_t(c) = W_H^\top W_V \phi(c)$ to refer to the inner product between the values and the embedding of all other tokens.

**Definition 1** (Memorization). *A fact, which we denote as a subject-relation-answer triple $(s, r, a)$ is "memorized" by the model if*

$$\sigma\left(\boldsymbol{v}(s)\right)_a = \sigma\left(W_H^\top W_V \phi(s)\right)_a > \delta_M \tag{17}$$

*where $\frac{1}{K_A} \ll \delta_M \leq 1$. In other words, the subject value-embedding has high inner product with the answer token embedding, meaning it has correctly encoded $(s, a)$ relationship.*

**Definition 2** (C Datapoints). *A Context Point $([c, s, r], a) \in \mathcal{D}_C$ where $c = a$ is one where*

$$\sigma\left(\boldsymbol{v}_0(c)\right)_c = \delta_C > \frac{3}{K_A - 1}, \sigma\left(\boldsymbol{v}_0(s)\right)_c = \sigma\left(\boldsymbol{v}_0(s)\right)_{c'} \quad \forall c' \in \mathcal{A} \tag{18}$$

*Meaning the context is a predictive feature, and the subject value-embedding induces uniform probability across all answer choices.*

**Definition 3** (C+S Datapoints). *A Context Point $([c, s, r], a) \in \mathcal{D}_{C+S}$ where $c = a$ is one where*

$$\sigma\left(\boldsymbol{v}_0(c)\right)_c = \delta_C, \sigma\left(\boldsymbol{v}_0(s)\right)_c = \delta_M > 2\delta_C \tag{19}$$

*So for a learned example, $\delta_M$ is more predictive than $\delta_C$, and $\delta_C$ is weakly predictive of the correct answer.*

**Assumption 1** (Non-Overlapping Subject-Answer). *We assume that any appearance of a subject $s_i \in \mathcal{D}$ is paired with a unique answer $a_i \in \mathcal{D}$. Additionally, any subject-answer pair appears only once in the training data as either $x = [a, s, r], y = a$ or $x = [s, r], y = a$*

### B.1.2 Token and Embedding Assumptions

We re-iterate key characteristics about the data. We consider a tokenizer with the set of all tokens equal to $\mathcal{T} = \mathcal{S} \cup \mathcal{A} \cup \{r\}$. The total size of $|\mathcal{S}| = K_S$ and $|\mathcal{A}| = K_A$ and $K_A > K_S$.

**Assumption 2** (Shared Direction). *We assume that the embeddings of all the subject tokens can be represented as the convex combination of with a shared direction $\theta_S$. Similarly, any context/answer token can be represented as the convex combination with a shared direction $\theta_C$. In other words,*

$$\forall s_i \in \mathcal{S}, \ \phi(s_i) = \sqrt{1/2}\tilde{s}_i + \sqrt{1/2}\theta_S \tag{20}$$

$$\forall a_i \in \mathcal{A}, \ \phi(c_i) = \sqrt{1/2}\tilde{a}_i + \sqrt{1/2}\theta_C \tag{21}$$

*where $\theta_S^\top \theta_C = 0$, $\theta_S \perp \mathcal{A}$, $\theta_C \perp \mathcal{S}$. Realistically, $\theta_S, \theta_C$ may encode some linguistic structure or meaning, e.g., the embedding of all country names may lie in the same direction.*

**Assumption 3** (Unitary Embeddings). *We assume that the embedding of all tokens is unitary $\|\phi(i)\|_2 = 1$. Specifically, $\|\theta_S\|_2, \|\theta_{C+S}\|_2, \|\phi(r)\|_2 = 1$ and $\|\tilde{c}_i\|_2, \|\tilde{s}_i\|_2 = 1 \forall s_i \in \mathcal{S}, c_i \in \mathcal{A}$*

**Assumption 4** (Orthogonal Embedding Constraints). *We assume the following:*

- $\phi(r) \perp \mathcal{S} \cup \mathcal{A}$

- $\tilde{s}_i \perp \tilde{s}_j, \quad \forall s_i, s_j \in \mathcal{S}$ where $i \neq j$

- $\tilde{c}_i \perp \tilde{c}_j, \quad \forall c_i, c_j \in \mathcal{A}$ where $i \neq j$

- $\tilde{s} \perp \tilde{c}, \quad \forall s \in \mathcal{S}, c \in \mathcal{A}$

### B.1.3 GENERAL PRETRAINED MODEL ASSUMPTIONS

**Assumption 5** (Pretrained Attention Weights Assumption). *We assume the following about $W_{QK}^0$ at timestep 0.*

- *For* C *and* C+S *points, we assume that the self-attention on the relation token $\sigma\left(\phi(r)^\top W_{QK}^{(0)}\phi(r)\right) = 0$ at the beginning of pretraining. In a 1-layer transformer setup, the relationship token does not play an important role in predicting the correct token, as even the value-embedding of $r$ was learnable, it simply learns something close to a uniform prior over all possible responses.*

- *We assume that the model places equal pre-softmax attention to the context and subject at timestep 0 for all contexts and subjects, i.e. $\forall c, c' \in \mathcal{A}$ and $s, s' \in \mathcal{S}$*

$$\phi(c)^\top W_{QK}^{(0)}\phi(r) = \phi(c')^\top W_{QK}^{(0)}\phi(r) = \phi(s)^\top W_{QK}^{(0)}\phi(r) = \phi(s')^\top W_{QK}^{(0)}\phi(r) \quad (22)$$

**Assumption 6** (Data Symmetry). *To ease our analysis, we assume the following symmetries of $W_V^0\phi(x)$. $\forall [c, s, r] \in \mathcal{D}$*

$$v_0(c', s) = v_0(c', c) = o_c \quad \forall c' \in \mathcal{A} \setminus \{c\}$$
$$v_0(r, c) = v_0(r, s) = v_0(r, r) = o_r \leq o_c$$
$$v_0(s', s) = v_0(s', c) = v_0(s', r) = 0 \quad \forall s' \in \mathcal{S}$$
$$v_0(c', r) = o_c \quad \forall c' \in \mathcal{A}$$

*where $o_c, o_r > 0$ are scalar values. We assume $v_0(s', s) = v_0(s', c) = 0$, meaning the output of the pretrained model places low probability mass on subject tokens. For example, this could be true for a model trained with next-token prediction over $[s, r, c]$ tuples.*

*Note that this implies that the quantity*

$$m = \langle \boldsymbol{v}_0(c) - \boldsymbol{v}_0(s), \boldsymbol{e}_c - \sigma(\boldsymbol{z}) \rangle$$

*where $\boldsymbol{z} = f_W([c, s, r])_r$ is equal across examples in $\mathcal{D}_C$, and similarly between any examples in $\mathcal{D}_{C+S}$. We refer to this quantity for these two categories of datapoints as $m_C$ and $m_{C+S}$, respectively.*

### B.1.4 PROOF OF PROPOSITION 1

**Proposition 1.** *When finetuning a one-layer transformer pretrained on $\mathcal{D}_{pre}$ with $W_V$ frozen over $\mathcal{D}^{SFT} = \mathcal{D}_C \cup \mathcal{D}_{C+S}$ with $|\mathcal{D}_C| \geq |\mathcal{D}_{C+S}|$, under Assumptions 1 to 6, there exists a learning rate $\eta^*$, such that the following holds true.*

- ***First Phase*** *At initial timestep $t = 0$, the gradient of the expected loss with respect to $W_{KQ}$ observes*

$$\theta_S^\top[-\nabla_{W_{KQ}}L(W^{(0)})]\phi(r) < 0, \quad \theta_C^\top[-\nabla_{W_{KQ}}L(W^{(0)})]\phi(r) > 0 \quad (23)$$

- ***Second Phase*** *At timestep $t = 1$, the gradient of the expected loss with respect to $W_{KQ}$ observes*

$$\theta_S^\top[-\nabla_{W_{KQ}}L(W^{(1)})]\phi(r) > 0, \quad \theta_C^\top[-\nabla_{W_{KQ}}L(W^{(1)})]\phi(r) < 0 \quad (24)$$

*Proof.* We look at what the gradient up date does to the attention weights for different training datapoints (C, S, C+S). We start by proving the following useful lemmas.

**Lemma 1.** *For a one-layer transformer, the gradient of the loss $\ell$ over example $\{[c, s, r], a\}$ with respect to the key-query weight matrix $W_{KQ}$ can be expressed as:*

$$-\nabla_{W_{KQ}}\ell(W, [c, s, r]) = \phi([c, s, r])[\mathrm{diag}(\boldsymbol{\sigma}_{csr}) - \boldsymbol{\sigma}_{csr}\boldsymbol{\sigma}_{csr}^\top]\phi([c, s, r])^\top W_V^\top W_H(e_c - \sigma(\boldsymbol{z}))\phi(r)^\top$$

*where $\boldsymbol{e}_c$ is an elementary vector and the softmax $\sigma$ is applied to each element of the model logits $\boldsymbol{z} = f_W([c, s, r])_r$ for the relation token $r$, and $\boldsymbol{\sigma}_{csr} = [\sigma_c, \sigma_s, \sigma_r]$ are the attention weights between the relation token and the context, subject, and relation tokens respectively.*

*Proof.* Rewriting Equation 4, we have:

$$\boldsymbol{z} = \sigma_c \boldsymbol{v}(c) + \sigma_s \boldsymbol{v}(s) + \sigma_r \boldsymbol{v}(r)$$

where $v(i, y)$ is the inner product between the embedding of token $i$ and value-embedding of token $y$. (Equation 16) and $\sigma_c, \sigma_s$ and $\sigma_r$ are the attention weights on context, subject and relation tokens respectively:

$$\sigma_c = \frac{\exp\left(\phi(c)^\top W_{KQ}\phi(r)\right)}{\sum_{y\in\{c,s,r\}}\exp\left(\phi(y)^\top W_{KQ}\phi(r)\right)},$$

$$\sigma_s = \frac{\exp\left(\phi(s)^\top W_{KQ}\phi(r)\right)}{\sum_{y\in\{c,s,r\}}\exp\left(\phi(y)^\top W_{KQ}\phi(r)\right)},$$

$$\sigma_r = \frac{\exp\left(\phi(r)^\top W_{KQ}\phi(r)\right)}{\sum_{y\in\{c,s,r\}}\exp\left(\phi(y)^\top W_{KQ}\phi(r)\right)}.$$

The gradient of $z_{ri}$ with respect to $W_{KQ}$ is given by:

$$\nabla_{W_{KQ}} z_{ri} = v(i,c)[\sigma_c(1-\sigma_c)\phi(c)\phi(r)^\top - \sigma_c\sigma_s\phi(s)\phi(r)^\top - \sigma_c\sigma_r\phi(r)\phi(r)^\top] \tag{25}$$

$$+v(i,s)[\sigma_s(1-\sigma_s)\phi(s)\phi(r)^\top - \sigma_s\sigma_c\phi(c)\phi(r)^\top - \sigma_s\sigma_r\phi(r)\phi(r)^\top] \tag{26}$$

$$+v(i,r)[\sigma_r(1-\sigma_r)\phi(r)\phi(r)^\top - \sigma_r\sigma_s\phi(s)\phi(r)^\top - \sigma_r\sigma_c\phi(c)\phi(r)^\top] \tag{27}$$

$$= \phi([c,s,r])[\mathrm{diag}(\boldsymbol{\sigma}_{csr}) - \boldsymbol{\sigma}_{csr}\boldsymbol{\sigma}_{csr}^\top]\phi([c,s,r])^\top W_V^\top \phi(i)\phi(r)^\top \tag{28}$$

Given the training loss $\ell(W, [c, x, r]) = -\log \sigma\left(f_W([c,x,r])_r\right)_c$, we have by chain rule:

$$-\nabla_{W_{KQ}}\ell(W, [c,s,r]) = \langle e_c - \sigma\left(\boldsymbol{z}\right), \nabla_{W_{KQ}}\boldsymbol{z}\rangle \tag{29}$$

$$= \phi([c,s,r])[\mathrm{diag}(\boldsymbol{\sigma}_{csr}) - \boldsymbol{\sigma}_{csr}\boldsymbol{\sigma}_{csr}^\top\phi([c,s,r])^\top W_V^\top W_H(e_c - \sigma\left(\boldsymbol{z}\right))\phi(r)^\top \tag{30}$$

$\square$

**Lemma 2.** *Note that*

$$-\theta_S^\top \nabla_{W_{KQ}}\ell(W, [c,s,r])\phi(r)$$

$$= \frac{1}{\sqrt{2}}(-\sigma_s\sigma_c\boldsymbol{v}_0(c) + (\sigma_s - \sigma_s^2)\boldsymbol{v}_0(s) - \sigma_s\sigma_r\boldsymbol{v}_0(r))^\top(e_c - \sigma\left(\boldsymbol{z}\right))$$

$$-\theta_C^\top \nabla_{W_{KQ}}\ell(W, [c,s,r])\phi(r)$$

$$= \frac{1}{\sqrt{2}}((\sigma_c - \sigma_c^2)\boldsymbol{v}_0(c) - \sigma_s\sigma_c\boldsymbol{v}_0(s) - \sigma_s\sigma_r\boldsymbol{v}_0(r))^\top(e_c - \sigma\left(\boldsymbol{z}\right))$$

*If* $\sigma_r = 0$, *the two quantities further simplify to* $\frac{\sigma_s\sigma_c}{\sqrt{2}}(\boldsymbol{v}_0(c) - \boldsymbol{v}_0(s))^\top(e_c - \sigma\left(\boldsymbol{z}\right))$ *and* $-\frac{\sigma_s\sigma_c}{\sqrt{2}}(\boldsymbol{v}_0(c) - \boldsymbol{v}_0(s))^\top(e_c - \sigma\left(\boldsymbol{z}\right))$, *respectively.*

*Proof.*

$$-\theta_S^\top \nabla_{W_{KQ}}\ell(W, [c,s,r])\phi(r) \tag{31}$$

$$= \theta_S^\top \phi([c,s,r])[\mathrm{diag}(\boldsymbol{\sigma}_{csr}) - \boldsymbol{\sigma}_{csr}\boldsymbol{\sigma}_{csr}^\top]\phi([c,s,r])^\top W_V^\top W_H(e_c - \sigma\left(\boldsymbol{z}\right))\underbrace{\|\phi(r)\|_2^2}_{=1} \tag{32}$$

$$= \frac{1}{\sqrt{2}}[-\sigma_s\sigma_c, \sigma_s - \sigma_s^2, -\sigma_s\sigma_r]^\top \phi([c,s,r])^\top W_V^\top W_H(e_c - \sigma\left(\boldsymbol{z}\right)) \tag{33}$$

$$= \frac{1}{\sqrt{2}}(-\sigma_s\sigma_c\boldsymbol{v}_0(c) + (\sigma_s - \sigma_s^2)\boldsymbol{v}_0(s) - \sigma_s\sigma_r\boldsymbol{v}_0(r))^\top(e_c - \sigma\left(\boldsymbol{z}\right)) \tag{34}$$

$\square$

**Lemma 3.** *For any example* $[c, s, r] \in \mathcal{D}_C$,

$$v_0(c, s) = o_c$$

$$v_0(c, c) = \log\left(\frac{\delta_C}{1 - \delta_C}\right) + \log\left((K_A - 1)\exp(o_c) + \exp(o_r) + K_S\right)$$

*For any example* $[c, s, r] \in \mathcal{D}_{C+S}$,

$$v_0(c, s) = \log\left(\frac{\delta_M}{1 - \delta_M}\right) + \log\left((K_A - 1)\exp(o_c) + \exp(o_r) + K_S\right)$$

$$v_0(c, c) = \log\left(\frac{\delta_C}{1 - \delta_C}\right) + \log\left((K_A - 1)\exp(o_c) + \exp(o_r) + K_S\right)$$

*Proof.* Recall from assumption 6, the following properties of any example in $\mathcal{D}$

$$v_0(c', s) = v_0(c', c) = o_c \quad \forall c' \in \mathcal{A} \setminus \{c\} \tag{35}$$

$$v_0(r, c) = v_0(r, s) = o_r \tag{36}$$

$$v_0(s', s) = v_0(s', c) = 0 \quad \forall s' \in \mathcal{S} \tag{37}$$

Take any example $[c, s, r] \in \mathcal{D}_C$. Recall that

$$\delta_C = \sigma\left(\boldsymbol{v}_0(c)\right)_c = \frac{\exp(v_0(c, c))}{(K_A - 1)\exp(o_c) + \exp(o_r) + \exp(v_0(c, c)) + K_S} \tag{38}$$

Thus

$$v_0(c, s) = o_c \tag{39}$$

$$v_0(c, c) = \log\left(\frac{\delta_C}{1 - \delta_C}\right) + \log\left((K_A - 1)\exp(o_c) + \exp(o_r) + K_S\right) \tag{40}$$

Similarly, take any example $[c, s, r] \in \mathcal{D}_{C+S}$. Recall that

$$\delta_M = \sigma\left(\boldsymbol{v}_0(s)\right)_c = \frac{\exp(v_0(c, s))}{(K_A - 1)\exp(o_c) + \exp(o_r) + \exp(v_0(c, s)) + K_S} \tag{41}$$

$$\delta_C = \sigma\left(\boldsymbol{v}_0(c)\right)_c = \frac{\exp(v_0(c, c))}{(K_A - 1)\exp(o_c) + \exp(o_r) + \exp(v_0(c, c)) + K_S} \tag{42}$$

Thus,

$$v_0(c, s) = \log\left(\frac{\delta_M}{1 - \delta_M}\right) + \log\left((K_A - 1)\exp(o_c) + \exp(o_r) + K_S\right) \tag{43}$$

$$v_0(c, c) = \log\left(\frac{\delta_C}{1 - \delta_C}\right) + \log\left((K_A - 1)\exp(o_c) + \exp(o_r) + K_S\right) \tag{44}$$

$\square$

**Lemma 4.** *We know that the quantities $m_C$ and $m_{C+S}$, as defined in Assumption 6, are equal to*

$$m_C = \lambda_C \left[\log\left(\frac{\delta_C}{1 - \delta_C}\right) + \log\left((K_A - 1)\exp(o_c) + \exp(o_r) + K_S\right) - o_c\right]$$

$$m_{C+S} = \lambda_{C+S} \left[\log\left(\frac{\delta_C}{1 - \delta_C}\right) - \log\left(\frac{\delta_M}{1 - \delta_M}\right)\right]$$

*where*

$$\lambda_C = \left(1 + \frac{\exp\left(\frac{1}{2}\log\left(\frac{\delta_C}{1 - \delta_C}\right) + \frac{1}{2}\log\left((K_A - 1)\exp(o_c) + \exp(o_r) + K_S\right) + \frac{1}{2}o_c\right)}{(K_A - 1)\exp(o_c) + \exp(o_r) + K_S}\right)^{-1} \tag{45}$$

$$\lambda_{C+S} = \left(1 + \frac{\exp\left(\frac{1}{2}\log\left(\frac{\delta_C}{1 - \delta_C}\right) + \frac{1}{2}\log\left(\frac{\delta_M}{1 - \delta_M}\right) + \log\left((K_A - 1)\exp(o_c) + \exp(o_r) + K_S\right)\right)}{(K_A - 1)\exp(o_c) + \exp(o_r) + K_S}\right)^{-1} \tag{46}$$

*Proof.* As per definition, $m_C$ and $m_{C+S}$ are equal to

$$= \langle \boldsymbol{v}_0(c) - \boldsymbol{v}_0(s), e_c - \sigma(\boldsymbol{z}) \rangle \tag{47}$$

$$= \left\langle \boldsymbol{v}_0(c) - \boldsymbol{v}_0(s), e_c - \sigma\left(\frac{1}{2}\boldsymbol{v}_0(c) + \frac{1}{2}\boldsymbol{v}_0(s)\right) \right\rangle \tag{48}$$

for any $[c, s, r] \in \mathcal{D}_\mathrm{C}$ and $\mathcal{D}_\mathrm{C+S}$, respectively.

We first calculate $m_C$. Let us simplify $\boldsymbol{v}_0(c) - \boldsymbol{v}_0(s)$. From Lemma 3 and Assumption 6, we know that for any $[c, s, r] \in \mathcal{D}_\mathrm{C}$

$$v_0(c, c) - v_0(c, s) \tag{49}$$

$$= \log\left(\frac{\delta_C}{1 - \delta_C}\right) + \log\left((K_A - 1)\exp(o_c) + \exp(o_r) + K_S\right) - o_c \tag{50}$$

and

$$v_0(s', c) - v_0(s', s) = 0 \quad \forall s' \in \mathcal{S} \tag{51}$$

$$v_0(c', c) - v_0(c', s) = o_c - o_c \quad \forall c' \in \mathcal{A} \setminus \{c\} \tag{52}$$

$$v_0(r, c) - v_0(r, s) = 0 \tag{53}$$

Therefore

$$m_C = (1 - \sigma(\boldsymbol{z})_c)\left[\log\left(\frac{\delta_C}{1 - \delta_C}\right) + \log\left((K_A - 1)\exp(o_c) + \exp(o_r) + K_S\right) - o_c\right] \tag{54}$$

for any $c' \in \mathcal{A} \setminus \{c\}$.

Next, we calculate $\sigma\left(\frac{1}{2}\boldsymbol{v}_0(c) + \frac{1}{2}\boldsymbol{v}_0(s)\right)_c$. Note that

$$\sum_{i \in \mathcal{T}} \exp(v_0(i)) \tag{55}$$

$$= \exp\left(\frac{1}{2}\log\left(\frac{\delta_C}{1 - \delta_C}\right) + \frac{1}{2}\log\left((K_A - 1)\exp(o_c) + \exp(o_r) + K_S\right) + \frac{1}{2}o_c\right) \tag{56}$$

$$+ (K_A - 1)\exp(o_c) + \exp(o_r) + K_S \tag{57}$$

and so

$$1 - \sigma(\boldsymbol{z})_c = 1 - \sigma\left(\frac{1}{2}\boldsymbol{v}_0(c) + \frac{1}{2}\boldsymbol{v}_0(s)\right)_c \tag{58}$$

$$= \left(1 + \frac{\exp\left(\frac{1}{2}\log\left(\frac{\delta_C}{1-\delta_C}\right) + \frac{1}{2}\log\left((K_A - 1)\exp(o_c) + \exp(o_r) + K_S\right) + \frac{1}{2}o_c\right)}{(K_A - 1)\exp(o_c) + \exp(o_r) + K_S}\right)^{-1} \tag{59}$$

Similarly, we compute $m_{C+S}$. From Lemma 3, we know

$$v_0(c, c) - v_0(c, s) \tag{60}$$

$$= \log\left(\frac{\delta_C}{1 - \delta_C}\right) + \log\left((K_A - 1)\exp(o_c) + \exp(o_r) + K_S\right) \tag{61}$$

$$- \log\left(\frac{\delta_M}{1 - \delta_M}\right) - \log\left((K_A - 1)\exp(o_c) + \exp(o_r) + K_S\right) \tag{62}$$

$$= \log\left(\frac{\delta_C}{1 - \delta_C}\right) - \log\left(\frac{\delta_M}{1 - \delta_M}\right) \tag{63}$$

And using Assumption 6, the other quantities in $\boldsymbol{v}_0(c) - \boldsymbol{v}_0(s)$ are the same as Equation 51, so

$$m_{C+S} = (1 - \sigma(\boldsymbol{z})_c)\left[\log\left(\frac{\delta_C}{1 - \delta_C}\right) - \log\left(\frac{\delta_M}{1 - \delta_M}\right)\right] \tag{64}$$

Next, we calculate $\sigma\left(\frac{1}{2}\boldsymbol{v}_0(c) + \frac{1}{2}\boldsymbol{v}_0(s)\right)_c$. Note that

$$\sum_{i \in \mathcal{T}} \exp(v_0(i)) \tag{65}$$

$$= \exp\left(\frac{1}{2}\log\left(\frac{\delta_C}{1-\delta_C}\right) + \frac{1}{2}\log\left(\frac{\delta_M}{1-\delta_M}\right) + \log\left((K_A - 1)\exp(o_c) + \exp(o_r) + K_S\right)\right) \tag{66}$$

$$+(K_A - 1)\exp\left(o_c\right) + \exp(o_r) + K_S \tag{67}$$

and so

$$1 - \sigma\left(\boldsymbol{z}\right)_c = 1 - \sigma\left(\frac{1}{2}\boldsymbol{v}_0(c) + \frac{1}{2}\boldsymbol{v}_0(s)\right)_c \tag{68}$$

$$= \left(1 + \frac{\exp\left(\frac{1}{2}\log\left(\frac{\delta_C}{1-\delta_C}\right) + \frac{1}{2}\log\left(\frac{\delta_M}{1-\delta_M}\right) + \log\left((K_A - 1)\exp(o_c) + \exp(o_r) + K_S\right)\right)}{(K_A - 1)\exp\left(o_c\right) + \exp(o_r) + K_S}\right)^{-1} \tag{69}$$

$\square$

**Lemma 5.** *The following is true,*

$$m_C > 0, m_{C+S} < 0$$

*Proof.* Refer to the form of $m_C$ and $m_{C+S}$ derived in Lemma 4. Note that $\lambda_{C+S}, \lambda_C > 0$ and since $\delta_M > \delta_C$ and $\frac{x}{1-x}$ is strictly increasing between 0 and 1,

$$\log\left(\frac{\delta_C}{1-\delta_C}\right) - \log\left(\frac{\delta_M}{1-\delta_M}\right) < 0 \tag{70}$$

Thus, $m_{C+S} < 0$. On the other hand, for $m_C > 0$ since

$$\log\left(\frac{\delta_C}{1-\delta_C}\right) + \log\left((K_A - 1)\exp(o_c) + \exp(o_r) + K_S\right) - o_c \tag{71}$$

$$\geq \log\left(\frac{1}{K_A - 1}\right) + \log\left((K_A - 1)\exp(o_c) + \exp(o_r) + K_S\right) - o_c \tag{72}$$

$$= \log\left(1 + \underbrace{\frac{\exp(o_r) + K_S}{(K_A - 1)\exp(o_c)}}_{>0}\right) \geq 0 \tag{73}$$

The first step follows by definition that $\delta_C > \frac{1}{K_A}$. $\square$

**Lemma 6.** *The following is true,*

$$|m_C| > |m_S|$$

*Proof.* From Lemma 4, note that

$$\frac{\lambda_C}{\lambda_{C+S}} = \frac{1 + \exp\left(\frac{1}{2}\log\left(\frac{\delta_C}{1-\delta_C}\right) + \frac{1}{2}\log\left(\frac{\delta_M}{1-\delta_M}\right)\right)}{1 + \exp\left(\frac{1}{2}\log\left(\frac{\delta_C}{1-\delta_C}\right) - \frac{1}{2}\log((K_A - 1)\exp(o_c) + \underbrace{\exp(o_r) + K_S}_{\geq 0}) + \frac{1}{2}o_c\right)} \tag{74}$$

$$\geq \frac{1 + \exp\left(\frac{1}{2}\log\left(\frac{\delta_C}{1-\delta_C}\right) + \frac{1}{2}\log\left(\frac{\delta_M}{1-\delta_M}\right)\right)}{1 + \exp\left(\frac{1}{2}\log\left(\frac{\delta_C}{1-\delta_C}\right) + \frac{1}{2}\log\left(\frac{1}{K_A - 1}\right)\right)} > 1 \tag{75}$$

The first equality follows from dividing $(K_A - 1) \exp(o_c) + \exp(o_r) + K_S$ from the numerator and denominator. Thus,

$$\frac{|m_C|}{|m_S|} = -\frac{m_C}{m_S} = \frac{\lambda_C}{\lambda_{C+S}} \cdot \frac{\log\left(\frac{\delta_C}{1-\delta_C}\right) + \log\left((K_A - 1)\exp(o_c) + \exp(o_r) + K_S\right) - o_c}{\log\left(\frac{\delta_M}{1-\delta_M}\right) - \log\left(\frac{\delta_C}{1-\delta_C}\right)} \quad (76)$$

$$\geq \frac{\exp\left(-\frac{1}{2}\log\left(\frac{\delta_C}{1-\delta_C}\right) + \frac{1}{2}\log\left(\frac{\delta_M}{1-\delta_M}\right)\right)}{\exp\left(-\frac{1}{2}\log\left(\frac{\delta_C}{1-\delta_C}\right) - \frac{1}{2}\log\left(K_A - 1\right)\right)} \cdot \frac{\log\left(\frac{\delta_C}{1-\delta_C}\right) + \log\left(K_A - 1\right)}{\log\left(\frac{\delta_M}{1-\delta_M}\right) - \log\left(\frac{\delta_C}{1-\delta_C}\right)} > 1 \quad (77)$$

For the last inequality we use the property that $\exp(\frac{1}{2}x) \geq x \;\forall x \in \mathbb{R}$ and $\exp(-\frac{1}{2}x) \leq x \;\forall x \in \mathbb{R}$ such that $x > 1$. So, $|m_C| \geq |m_S|$. $\qquad \square$

**Proof of First Phase**  At the beginning of training, we assumed in Assumption 5 that the attention weights between the context and subject is equal at the beginning of training for all datapoints $x \in \mathcal{D}^{SFT}$, i.e., $\sigma_s^0 = \sigma_c^0 = 1/2$ and $\sigma_r^0 = 0$.

Using Lemma 2, it follows that

$$-\theta_C^\top \nabla_{W_{KQ}} \ell(W^{(0)}, [c, s, r])\theta(r) = \frac{1}{4\sqrt{2}}(\boldsymbol{v}_0(c) - \boldsymbol{v}_0(s))^\top(e_c - \sigma(\boldsymbol{z})) \quad (78)$$

which equals $\frac{1}{4\sqrt{2}} m_C$ for $[c, s, r] \in \mathcal{D}_C$ and $\frac{1}{4\sqrt{2}} m_{C+S}$ for $[c, s, r] \in \mathcal{D}_{C+S}$.

Using Lemma 5, and Lemma 4 it directly follows that

$$\theta_C^\top[-\nabla_{W_{KQ}} L(W^)]\theta_r = \frac{1}{8\sqrt{2}} m_C + \frac{1}{8\sqrt{2}} m_{C+S} > 0 \quad (79)$$

Since $\theta_S^\top[-\nabla_{W_{KQ}} L(W^)]\theta_r = -\theta_C^\top[-\nabla_{W_{KQ}} L(W^)]\theta_r$, it directly follows that $\theta_S^\top[-\nabla_{W_{KQ}} L(W^)]\theta_r < 0$. This completes the proof for the first phase.

**Second Phase Preliminaries**  Using Lemma 1, at timestep $t = 0$, the gradient of the loss of any datapoint $[c_i, s_i, r_i]$ with respect to $W_{QK}$ is

$$-\nabla_{W_{KQ}} \ell(W, [c, s, r]) \quad (80)$$

$$= \phi([c, s, r])[\text{diag}(\boldsymbol{\sigma}_{csr}) - \boldsymbol{\sigma}_{csr}\boldsymbol{\sigma}_{csr}^\top] \underbrace{\phi([c, s, r])^\top W_V^\top W_H}_{[\boldsymbol{v}(c), \boldsymbol{v}(s), \boldsymbol{v}(r)]^\top}(e_c - \sigma(\boldsymbol{z}))\phi(r)^\top \quad (81)$$

$$= \frac{1}{4}\langle \boldsymbol{v}(c) - \boldsymbol{v}(s), e_c - \sigma(\boldsymbol{z})\rangle(\phi(c) - \phi(s))\phi(r)^\top \quad (82)$$

where $\boldsymbol{z} = \frac{1}{2}\boldsymbol{v}(c) + \frac{1}{2}\boldsymbol{v}(s)$ and $\boldsymbol{\sigma}_{csr} = [\frac{1}{2}, \frac{1}{2}, 0]$

Consider taking a full batch gradient update step

$$W_{KQ}^1 = W_{KQ}^0 - \frac{\eta}{n}\sum_{i=1=}^n \nabla_{W_{KQ}} \ell(W, [c_i, s_i, r]),$$

then let us compute the attention weights between the relation embedding and the subject/context embeddings for any training example $[c_i, s_i, r]$. First, note that

$$\phi(c_i)^\top \left( -\sum_{j=1}^n \nabla_{W_{KQ}} \ell(W, [c_j, s_j, r]) \right) \phi(r) \tag{83}$$

$$= \frac{1}{4} \sum_{j=1}^n \langle \boldsymbol{v}(c_j) - \boldsymbol{v}(s_j), \boldsymbol{e}_{c_j} - \sigma(\boldsymbol{z}_{rj}) \rangle \|\phi(r)\| \langle \phi(c_i), \phi(c_j) - \phi(s_j) \rangle \tag{84}$$

$$= \frac{1}{4} \left[ m_C \sum_{j=1}^{n/2} \langle \phi(c_i), \phi(c_j) - \phi(s_j) \rangle + m_{C+S} \sum_{j=n/2+1}^n \langle \phi(c_i), \phi(c_j) - \phi(s_j) \rangle \right] \tag{85}$$

$$= \frac{1}{8} [m_C \sum_{j=1}^n (1 + \mathbb{1}[i=j]) + m_{C+S} \sum_{j=n/2+1}^n (1 + \mathbb{1}[i=j]) )] \tag{86}$$

where $n = |\mathcal{D}|$ and we refer to all examples in $\mathcal{D}_C$ as $[c_j, s_j, r]_{j=1}^{n/2}$ and in $\mathcal{D}_{C+S}$ as $[c_j, s_j, r]_{j=n/2+1}^n$. The last step follows from assumption 4. Furthermore, one can easily calculate that

$$\phi(s_i)^\top \left( -\sum_{j=1}^n \nabla_{W_{KQ}} \ell(W, [c_j, s_j, r]) \right) \phi(r) = \phi(c_i)^\top \left( \sum_{j=1}^n \nabla_{W_{KQ}} \ell(W, [c_j, s_j, r]) \right) \phi(r) \tag{87}$$

So for any datapoint $[c_i, s_i, r] \in \mathcal{D}_C$,

$$\phi(c_i)^\top W_{KQ}^1 \phi(r) = \phi(c_i)^\top W_{KQ}^0 \phi(r) + \frac{\eta}{16} \left[ m_C \left( \frac{n+2}{n} \right) + m_{C+S} \right] \tag{88}$$

$$\phi(s_i)^\top W_{KQ}^1 \phi(r) = \phi(s_i)^\top W_{KQ}^0 \phi(r) - \frac{\eta}{16} \left[ m_C \left( \frac{n+2}{n} \right) + m_{C+S} \right] \tag{89}$$

and similarly, for any datapoint $[c_i, s_i, r] \in \mathcal{D}_{C+S}$,

$$\phi(c_i)^\top W_{KQ}^1 \phi(r) = \phi(c_i)^\top W_{KQ}^0 \phi(r) + \frac{\eta}{16} \left[ m_C + m_{C+S} \left( \frac{n+2}{n} \right) \right] \tag{90}$$

$$\phi(s_i)^\top W_{KQ}^1 \phi(r) = \phi(s_i)^\top W_{KQ}^0 \phi(r) - \frac{\eta}{16} \left[ m_C + m_{C+S} \left( \frac{n+2}{n} \right) \right] \tag{91}$$

Going back to Equation 88 and 90, note that

$$A_1 = \left( \frac{n+2}{n} \right) m_C + m_{C+S} > \frac{2}{n} m_C > 0 \tag{92}$$

$$A_2 = m_C + \left( \frac{n+2}{n} \right) m_{C+S} > \frac{2}{n} m_{C+S} \tag{93}$$

$$|A_1| > |A_2| \tag{94}$$

Thus, the attention to context strictly increases from $t = 0$ to $t = 1$ for $\mathcal{D}_C$ points, while for $n > 2 \frac{|m_{C+S}|}{|m_C| - |m_{C+S}|}$, the attention to context also increases for $\mathcal{D}_{C+S}$ by a smaller degree. Specifically, using Assumption 5, it easily follows that

$$\sigma \left( \phi(c)^\top W_{KQ}^1 \phi(r) \right) = \frac{1}{1 + \exp(-\frac{\eta}{8} A_1)} \quad \forall [c, s, r] \in \mathcal{D}_C \tag{95}$$

$$\sigma \left( \phi(s)^\top W_{KQ}^1 \phi(r) \right) = \frac{1}{1 + \exp(\frac{\eta}{8} A_1)} \quad \forall [c, s, r] \in \mathcal{D}_C \tag{96}$$

$$\sigma \left( \phi(c)^\top W_{KQ}^1 \phi(r) \right) = \frac{1}{1 + \exp(-\frac{\eta}{8} A_2)} \quad \forall [c, s, r] \in \mathcal{D}_{C+S} \tag{97}$$

$$\sigma \left( \phi(s)^\top W_{KQ}^1 \phi(r) \right) = \frac{1}{1 + \exp(\frac{\eta}{8} A_2)} \quad \forall [c, s, r] \in \mathcal{D}_{C+S} \tag{98}$$

**Lemma 7.** *At timestep $t = 0$, for any learning rate $\eta \in (0, \infty)$, the prediction towards the answer $\sigma\left(\mathbf{z}^1\right)_c$ increases monotonically with $\eta$ for $\mathcal{D}_C$ examples while decreasing monotonically for $\mathcal{D}_{C+S}$ examples.*

*Proof.* Setting $\sigma_c^1 = \sigma\left(\phi(c)^\top W_{KQ}^1 \phi(r)\right)$, note that for any $[c, s, r] \in \mathcal{D}$

$$\sigma\left(\mathbf{z}^1\right)_c = \frac{\exp(\sigma_c^1 v_0(c, c) + (1 - \sigma_c^1) v_0(c, s))}{\exp(\sigma_c^1 v_0(c, c) + (1 - \sigma_c^1) v_0(c, s)) + (K_A - 1)\exp(o_c) + \exp(o_r) + K_S} \tag{99}$$

$$\tag{100}$$

For examples in $\mathcal{D}_C$, $v_0(c, c) > v_0(c, s)$ by construction and $\sigma_c^1$ increases monotonically with $\eta$, so $\exp(\sigma_c^1 v_0(c, c) + (1 - \sigma_c^1) v_0(c, s))$ increases monotonically. This implies $\sigma(\mathbf{z}^1)_c$ increases monotonically. On the other hand, for examples in $\mathcal{D}_{C+S}$, $v_0(c, c) < v_0(c, s)$ by construction and $\sigma_c^1$ increases monotonically with $\eta$, so $\exp(\sigma_c^1 v_0(c, c) + (1 - \sigma_c^1) v_0(c, s))$ decreases monotonically. This implies $\sigma(\mathbf{z}^1)_c$ decreases monotonically.

$\square$

**Second Phase**   Now, we calculate the gradient of $W_{KQ}$ at timestep $t = 1$. Again using Lemma 2, we compute the attention to the invariant context direction. Note that $\forall [c, s, r] \in \mathcal{D}_C$

$$-\theta_C \nabla_{W_{KQ}} \ell(W^1, [c, s, r]) \phi(r) \tag{101}$$

$$= \frac{\exp(\frac{\eta}{8} A_1)}{\sqrt{2}(1 + \exp(\frac{\eta}{8} A_1))^2} (\mathbf{v}_0(c) - \mathbf{v}_0(s))^\top (e_c - \sigma(\mathbf{z}_C^1)) \tag{102}$$

$$= \frac{\exp(\frac{\eta}{8} A_1)(1 - \sigma\left(\mathbf{z}_C^1\right)_c)}{\sqrt{2}(1 + \exp(\frac{\eta}{8} A_1))^2} \left[\log\left(\frac{\delta_C}{1 - \delta_C}\right) + \log\left((K_A - 1)\exp(o_c) + \exp(o_r) + K_S\right) - o_c\right] \tag{103}$$

$$\leq \frac{\exp(\frac{\eta}{8} A_1)(1 - \frac{1}{K_A})}{\sqrt{2}(1 + \exp(\frac{\eta}{8} A_1))^2} \left[\log\left(\frac{\delta_C}{1 - \delta_C}\right) + \log\left(K_A\right)\right] \tag{104}$$

Similarly, $\forall [c, s, r] \in \mathcal{D}_{C+S}$

$$-\theta_C \nabla_{W_{KQ}} \ell(W^1, [c, s, r]) \phi(r) = \frac{\exp(\frac{\eta}{8} A_2)(1 - \sigma\left(\mathbf{z}_{C+S}^1\right)_c)}{\sqrt{2}(1 + \exp(\frac{\eta}{8} A_2))^2} \left[\log\left(\frac{\delta_C}{1 - \delta_C}\right) - \log\left(\frac{\delta_M}{1 - \delta_M}\right)\right] \tag{105}$$

$$\leq \frac{\exp(\frac{\eta}{8} A_2)(1 - \delta_M)}{\sqrt{2}(1 + \exp(\frac{\eta}{8} A_2))^2} \left[\log\left(\frac{\delta_C}{1 - \delta_C}\right) - \log\left(\frac{\delta_M}{1 - \delta_M}\right)\right] \tag{106}$$

We argue there exists a finite $\eta^*$ such that

$$\frac{\exp(\frac{\eta}{8} A_2)}{(1 + \exp(\frac{\eta}{8} A_2))^2} \cdot \frac{(1 + \exp(\frac{\eta}{8} A_1))^2}{\exp(\frac{\eta}{8} A_1)} \geq \frac{1 - \frac{1}{K_A}}{1 - \delta_M} \cdot \underbrace{\frac{\log\left(\frac{\delta_C}{1 - \delta_C}\right) + \log\left(K_A\right)}{\log\left(\frac{\delta_M}{1 - \delta_M}\right) - \log\left(\frac{\delta_C}{1 - \delta_C}\right)}}_{>1} \tag{107}$$

since

$$\lim_{\eta \to \infty} \frac{\exp(\frac{\eta}{8} A_2)}{(1 + \exp(\frac{\eta}{8} A_2))^2} \cdot \frac{(1 + \exp(\frac{\eta}{8} A_1))^2}{\exp(\frac{\eta}{8} A_1)} \tag{108}$$

$$= \lim_{\eta \to \infty} \frac{(1 + \exp(\frac{\eta}{8} A_1))(1 + \exp(-\frac{\eta}{8} A_1))}{(1 + \exp(\frac{\eta}{8} A_2))(1 + \exp(-\frac{\eta}{8} A_2))} \tag{109}$$

$$= \lim_{\eta \to \infty} \frac{1 + \exp(\frac{\eta}{8} A_1)}{1 + \exp(\frac{\eta}{8} A_2))} = \infty \tag{110}$$

where the last line follows because we know from Lemma 6 $A_1 > A_2$.

Setting $\eta = \eta^*$, note that the attention weight of the average gradient to the invariant context direction is negative.

$$\theta_C^\top \left[ -\frac{1}{n} \sum_{[c,s,r] \in \mathcal{D}} \nabla_{W_{KQ}} \ell(W^1, [c,s,r]) \right] \phi(r) \tag{111}$$

$$\leq \frac{\exp(\frac{\eta^*}{8} A_1)(1 - \frac{1}{K_A})}{2\sqrt{2}(1 + \exp(\frac{\eta^*}{8} A_1))^2} \left[ \log\left(\frac{\delta_C}{1 - \delta_C}\right) + \log(K_A) \right] \tag{112}$$

$$\qquad + \frac{\exp(\frac{\eta^*}{8} A_2)(1 - \delta_M)}{2\sqrt{2}(1 + \exp(\frac{\eta^*}{8} A_2))^2} \left[ \log\left(\frac{\delta_C}{1 - \delta_C}\right) - \log\left(\frac{\delta_M}{1 - \delta_M}\right) \right]$$

$$< 0 \tag{113}$$

$\square$

## B.2 Proof of Proposition 2

**Proposition 2** (More Attention to Subject with S Points). *Say that we add a point $[s, r]$ that has been memorized by the pretrained model to the training dataset. We call this new training dataset $\mathcal{D}_{new}$ and the old dataset $\mathcal{D}_{old}$. Under assumptions listed in Appendix B.1. At timestep $t = 0$*

$$\theta_S^\top[-\nabla_{W_{KQ}} L(W^{(0)}, \mathcal{D}_{new})]\phi(r) > \theta_S^\top[-\nabla_{W_{KQ}} L(W^{(0)}, \mathcal{D}_{old})]\phi(r) \tag{114}$$

$$\theta_C^\top[-\nabla_{W_{KQ}} L(W^{(0)}, \mathcal{D}_{new})]\phi(r) = \theta_C^\top[-\nabla_{W_{KQ}} L(W^{(0)}, \mathcal{D}_{old})]\phi(r) \tag{115}$$

*Proof.* Using Lemma 1, it follows that for any memorized point $[s, r] \in \mathcal{D}_S$

$$\theta_S^\top[-\nabla_{W_{KQ}} \ell(W, [s, r])]\phi(r) \tag{116}$$

$$= \frac{1}{\sqrt{2}} \sigma_s \sigma_r (\boldsymbol{v}_0(s) - \boldsymbol{v}_0(r))^\top (\boldsymbol{e}_c - \sigma(\boldsymbol{z})) \tag{117}$$

Using Assumption 6, note that

$$v(s, s) - v(s, r) = 0 \tag{118}$$

$$v(c', s) - v(c', r) = o_c - o_c = 0 \quad \forall c' \in \mathcal{C}/\{a\} \tag{119}$$

$$v(a, s) - v(a, r) > 0 \tag{120}$$

Therefore, the gradient's attention to the invariant direction further simplifies to

$$= \frac{1}{\sqrt{2}}(v(a, s) - v(a, r))(1 - \sigma(f_W([s, r])_r)_a) > 0 \tag{121}$$

Since $\theta_S^\top[-\nabla_{W_{KQ}} L(W^{(0)}, \mathcal{D}_{old})]\phi(r) < 0$, then $\theta_S^\top[-\nabla_{W_{KQ}} L(W^{(t)}, \mathcal{D}_{new})]\phi(r) > \theta_S^\top[-\nabla_{W_{KQ}} L(W^{(0)}, \mathcal{D}_{old})]\phi(r)$.

On the other hand, since $\theta_C$ is orthogonal by construction to any $\phi(s)$ for $s \in \mathcal{S}$ and $\phi(r)$,

$$\theta_C^\top[-\nabla_{W_{KQ}} \ell(W, [s, r])]\phi(r) = 0 \tag{122}$$

This completes our proof. $\square$

## B.3 PROOF OF PROPOSITION 3

**Proposition 3** (Fact Memorization). *Under Assumptions in Appendix B.1, for any example $[c, s, r] \in \mathcal{D}_C$, after the gradient step at timestep $t = 0$, the value embedding of the subject token is more predictive of the label $c$.*

$$\sigma \left( W_H^\top W_V^{(1)} \phi(s) \right)_c - \sigma \left( W_H^\top W_V^{(0)} \phi(s) \right)_c > 0 \tag{123}$$

*Proof.*

$$-\nabla_{W_V} L(W) = \frac{1}{n} \sum_{i=1}^{n} \langle e_{c_i} - \sigma(\boldsymbol{z}_i), \nabla_{W_V} \boldsymbol{z}_i \rangle \tag{124}$$

$$= \frac{1}{n} \sum_{i=1}^{n} W_H(e_{c_i} - \sigma(\boldsymbol{z}_i))[\sigma_{c_i} \phi(c_i) + \sigma_{s_i} \phi(s_i) + \sigma_r \phi(r)]^\top \tag{125}$$

For $[c_j, s_j, r_j] \in \mathcal{D}_C$,

$$v_{t+1}(c_j, s_j) - v_t(c_j, s_j) = -\eta \phi(c_j)^\top \nabla_{W_V} L(W) \phi(s_j) \tag{126}$$

$$= \frac{\eta}{n} \sum_{i=1}^{n} \frac{(1 + \mathbb{1}[i = j])}{4} (e_{c_i} - \sigma(\boldsymbol{z}_i))^\top W_H^\top \phi(c_j) \tag{127}$$

$$= \frac{\eta}{n} \sum_{i=1}^{n} \frac{(1 + \mathbb{1}[i = j])}{4} \left( \frac{1 + \mathbb{1}[i = j]}{2}(1 - \sigma(\boldsymbol{z}_i)_{c_i}) - \frac{|\mathcal{C}| + 1 - 2\mathbb{1}[i = j]}{2} \sigma(\boldsymbol{z}_i)_{c_k} \right) \text{ where } c_k \neq c_i \tag{128}$$

$$= \frac{\eta}{8n} \left( 2(1 - \delta_C) + \sum_{i \neq j} |\mathcal{S}| \sigma(\boldsymbol{z}_i)_s + \sum_{i \neq j} \sigma(\boldsymbol{z}_i)_r - 2 \sum_{i \neq j} \sigma(\boldsymbol{z}_i)_{c_j} + 2|\mathcal{S}| \sigma(\boldsymbol{z}_j)_s + 2\sigma(\boldsymbol{z}_j)_r \right) \tag{129}$$

where we use the fact that $\sigma_s = 0.5$ for all examples at timestep 0. Similarly,

$$\forall k \neq j, \quad v_{t+1}(c_k, s_j) - v_t(c_k, s_j) \tag{130}$$

$$= \frac{\eta}{n} \sum_{i=1}^{n} \frac{(1 + \mathbb{1}[i = j])}{4} \left( \frac{1 + \mathbb{1}[i = k]}{2}(1 - \sigma(\boldsymbol{z})_{c_i}) - \frac{|\mathcal{C}| + 1 - 2\mathbb{1}[i = k]}{2} \sigma(\boldsymbol{z})_{c_{k'}} \right) \text{ where } c_k \neq c_i \tag{131}$$

$$= \frac{\eta}{8n} \left( (1 - \delta_C) + \sum_{i=1}^{n} |\mathcal{S}| \sigma(\boldsymbol{z}_i)_s + \sum_{i=1}^{n} \sigma(\boldsymbol{z}_i)_r - 2 \sum_{i \neq k} \sigma(\boldsymbol{z}_i)_{c_k} + |\mathcal{S}| \sigma(\boldsymbol{z}_j)_s + \sigma(\boldsymbol{z}_j)_r - 2\sigma(\boldsymbol{z}_j)_{c_k} \right) \tag{132}$$

$$\forall c' \notin \mathcal{D}, \quad v_{t+1}(c', s_j) - v_t(c', s_j) \text{where } c' \notin \mathcal{D}, c_{k'} \neq c_i \tag{133}$$

$$= \frac{\eta}{n} \sum_{i=1}^{n} \frac{(1 + \mathbb{1}[i = j])}{4} \left( \frac{1}{2}(1 - \sigma(\boldsymbol{z})_{c_i}) - \frac{|\mathcal{C}| - 1}{2} \sigma(\boldsymbol{z})_{c_k} \right) \tag{134}$$

$$= \frac{\eta}{8n} \left( \sum_{i=1}^{n} |\mathcal{S}| \sigma(\boldsymbol{z}_i)_s + \sum_{i=1}^{n} \sigma(\boldsymbol{z}_i)_r - 2 \sum_{i=1}^{n} \sigma(\boldsymbol{z}_i)_{c_k} + |\mathcal{S}| \sigma(\boldsymbol{z}_j)_s + \sigma(\boldsymbol{z}_j)_r - 2\sigma(\boldsymbol{z}_j)_{c_k} \right) \tag{135}$$

$$v_{t+1}(s, s_j) - v_t(s, s_j) = -\eta |\mathcal{S}| \left( \frac{\sigma(\boldsymbol{z}_C)_s(n + 2)}{8n} + \frac{\sigma(\boldsymbol{z}_{C+S})_s}{8} \right) \tag{136}$$

$$v_{t+1}(r, s_j) - v_t(r, s_j) = -\eta \left( \frac{\sigma(\boldsymbol{z}_C)_r(n + 2)}{8n} + \frac{\sigma(\boldsymbol{z}_{C+S})_r}{8} \right) \tag{137}$$

We use $\boldsymbol{\sigma}(z_C)_x, \sigma(\boldsymbol{z}_{C+S})_x$ to denote the value of these quantities for any example $[c, s, r] \in \mathcal{D}_C$ and $\mathcal{D}_{C+S}$, respectively. By the data symmetry assumption (6), these quantities are equal within each

category of examples. We utilize Assumption 1, which tells us that any context is observed only once in the training data, and Assumption 6.

Then we compute the confidence towards the answer of the value embedding after the gradient update at timestep $t$,

$$\sigma\left(\boldsymbol{v}_{t+1}(s_j)\right)_{c_j} = \tag{138}$$

$$\left(1 + \frac{(n-1)\exp(v_{t+1}(c_k, s_j)) + (|\mathcal{C}| - n)\exp(v_{t+1}(c', s_j)) + \sum_{s \in \mathcal{S}} v_{t+1}(s, s_j) + v_{t+1}(r, s_j)}{\exp(v_{t+1}(c_j, s_j))}\right)^{-1} \tag{139}$$

where $k \neq j$ and $c' \notin \mathcal{D}$.

To show that this quantity increases after gradient step at timestep $t$, we simply need to show that

$$\forall k \in [n] \setminus i, \quad \frac{\exp\left(v_{t+1}(c_k, s_j) - v_t(c_k, s_j)\right)}{\exp\left(v_{t+1}(c_j, s_j) - v_t(c_j, s_j)\right)} < 1 \tag{140}$$

$$\forall c' \in \mathcal{C} \setminus \mathcal{D}, \quad \frac{\exp\left(v_{t+1}(c', s_j) - v_t(c', s_j)\right)}{\exp\left(v_{t+1}(c_j, s_j) - v_t(c_j, s_j)\right)} < 1 \tag{141}$$

$$\forall s \in \mathcal{S}, \quad \frac{\exp\left(v_{t+1}(s, s_j) - v_t(s, s_j)\right)}{\exp\left(v_{t+1}(c_j, s_j) - v_t(c_j, s_j)\right)} < 1 \tag{142}$$

$$\frac{\exp\left(v_{t+1}(r, s_j) - v_t(r, s_j)\right)}{\exp\left(v_{t+1}(c_j, s_j) - v_t(c_j, s_j)\right)} < 1 \tag{143}$$

This is equivalent to showing that

$$v_{t+1}(c_k, s_j) - v_t(c_k, s_j) - v_{t+1}(c_j, s_j) + v_t(c_j, s_j) = \frac{\eta}{8n}\left(-(1 - \delta_C) - 2\sigma(\boldsymbol{z}_j)_{c_j}\right) < 0 \tag{144}$$

$$v_{t+1}(c', s_j) - v_t(c', s_j) - v_{t+1}(c_j, s_j) + v_t(c_j, s_j) = \frac{\eta}{8n}(-2(1 - \delta_C) - 4\sigma(\boldsymbol{z}_j)_{c_k} < 0 \tag{145}$$

$$v_{t+1}(s', s_j) - v_t(s', s_j) - v_{t+1}(c_j, s_j) + v_t(c_j, s_j) \leq -2\eta\,|\mathcal{S}|\left(\frac{\sigma(\boldsymbol{z}_\text{C})_s(n+2)}{8n} + \frac{\sigma(\boldsymbol{z}_\text{C+S})_s}{8}\right) < 0 \tag{146}$$

$$v_{t+1}(r, s_j) - v_t(r, s_j) - v_{t+1}(c_j, s_j) + v_t(c_j, s_j) \leq -2\eta\left(\frac{\sigma(\boldsymbol{z}_\text{C})_r(n+2)}{8n} + \frac{\sigma(\boldsymbol{z}_\text{C+S})_r}{8}\right) \leq 0 \tag{147}$$

This completes our proof. $\qquad\square$

### B.4 PROOF OF THEOREM 1

**Theorem 1** (Test-Time Dynamic). *Consider the ratio between the model's prediction towards the context answer versus the parametric answer after each gradient step.*

$$M_C^{(t)} = \frac{\sigma(\boldsymbol{z}^{(t)})_c}{(\sigma(\boldsymbol{z}^{(t)})_c + \sigma(\boldsymbol{z}^{(t)})_a)} \tag{148}$$

*where $\boldsymbol{z}^{(t)} = f_{W^{(t)}}([c, s, r])_r$ denotes the model's unnormalized next-token probabilities at timestep $t$. Under the setting described in Proposition 1, for a counterfactual test example $[c, s, r]$ that was memorized at pretraining and $c \notin \mathcal{D}$, it directly follows that*

$$M_C^{(1)} > M_C^{(0)}, M_C^{(1)} > M_C^{(2)} \tag{149}$$

*Proof.* We now consider a counterfactual datapoint $[c, s, r]$ where the answer $a \neq c$, and the answer was memorized by the model at pretraining.

Note that for all $[c', s', r] \in \mathcal{D}$

$$\phi([c, s, r])^\top \phi([c', s', r]) = \text{diag}([1/2, 1/2, 1]) \tag{150}$$

Then note that, at any timestep,

$$-\phi(c)^\top \nabla_{W_{KQ}} \ell(W, [c', s', r])\phi(r) = -\frac{1}{\sqrt{2}}\theta_C^\top \nabla_{W_{KQ}} \ell(W, [c', s', r])\phi(r) \tag{151}$$

$$-\phi(s)^\top \nabla_{W_{KQ}} \ell(W, [c', s', r])\phi(r) = -\frac{1}{\sqrt{2}}\theta_S^\top \nabla_{W_{KQ}} \ell(W, [c', s', r])\phi(r) \tag{152}$$

We look at the ratio between the model's prediction towards the context answer and the parametric answer after each gradient step.

$$\frac{\sigma(z_r)_c}{(\sigma(z_r)_c + \sigma(z_r)_a)} \tag{153}$$

$$\frac{\sigma(z_r^1)_c}{(\sigma(z_r^1)_c + \sigma(z_r^1)_a)} > \frac{\sigma(z_r^0)_c}{(\sigma(z_r^0)_c + \sigma(z_r^0)_a)} \tag{154}$$

$$\frac{\sigma(z_r^1)_c}{(\sigma(z_r^1)_c + \sigma(z_r^1)_a)} > \frac{\sigma(z_r^2)_c}{(\sigma(z_r^2)_c + \sigma(z_r^2)_a)} \tag{155}$$

$$\tag{156}$$

By definition, we know

$$v(c, c) = \log\left(\frac{\delta_C}{1 - \delta_C}\right) + \log((K_A - 1)\exp(o_c) + \exp(o_r) + K_S) \tag{157}$$

$$v(a, s) = \log\left(\frac{\delta_M}{1 - \delta_M}\right) + \log((K_A - 1)\exp(o_c) + \exp(o_r) + K_S) \tag{158}$$

$$v(c', s) = o_c \tag{159}$$

$$v(c', c) = o_c \quad \forall c' \in \mathcal{A} \setminus \{c\} \tag{160}$$

$$v(r, c) = v(r, s) = o_r \tag{161}$$

$$\tag{162}$$

and

$$\frac{\sigma(z_r^1)_c}{(\sigma(z_r^1)_a + \sigma(z_r^1)_c)} = \tag{163}$$

$$\left(1 + \frac{\exp((1 - \sigma_c)\log\left(\frac{\delta_M}{1 - \delta_M}\right) + (1 - \sigma_c)\log((K_A - 1)\exp(o_c) + \exp(o_r) + K_S) + \sigma_c o_c)}{\exp(\sigma_c \log\left(\frac{\delta_C}{1 - \delta_C}\right) + \sigma_c \log((K_A - 1)\exp(o_c) + \exp(o_r) + K_S) + (1 - \sigma_c)o_c)}\right)^{-1} \tag{164}$$

$$= \left(1 + \underbrace{\frac{\exp\left((1 - \sigma_c)\log\left(\frac{\delta_M}{1 - \delta_M}\right) - \sigma_c \log\left(\frac{\delta_C}{1 - \delta_C}\right)\right)}{\exp((2\sigma_c - 1)\log((K_A - 1) + (\exp(o_r) + K_S)/\exp(o_c))}}_{=X}\right)^{-1} \tag{165}$$

We track the value of $\sigma_c$ over the timesteps. Note that since $\log\left(\frac{\delta_M}{1 - \delta_M}\right) > \log\left(\frac{\delta_C}{1 - \delta_C}\right)$ by construction, $X$ monotonically decreases with respect to $\delta_C$, which forces $\frac{\sigma(z_r^1)_c}{(\sigma(z_r^1)_a + \sigma(z_r^1)_c)}$ to strictly increase. Note that at timestep $t = 1$, $\sigma_c$ is largest, meaning $\frac{\sigma(z_r^1)_c}{(\sigma(z_r^1)_a + \sigma(z_r^1)_c)}$ is largest at timestep $t = 1$. This completes our proof. $\qquad\square$

