# OpenReview forum: "Context-Parametric Inversion: Why Instruction Finetuning May Not Actually Improve Context Reliance"
_ICLR.cc/2025/Conference — ICLR 2025 Oral_

### Official Review · Reviewer_DZJy · 2024-10-31

**Soundness:** 4
**Presentation:** 4
**Contribution:** 4
**Rating:** 8
**Confidence:** 4

**Summary:**

This paper reveals a counterintuitive phenomenon where LLMs become less reliant on contextual information despite being finetuned to enhance their ability to follow instructions and context. Through empirical studies and theoretical analysis, the authors identify the cause of this "context-parametric inversion" and propose mitigation strategies, offering valuable insights into improving LLM performance in context-dependent tasks.

**Strengths:**

This is a very interesting paper that identifies an obvious flaw in instruction fine-tuning and gives preliminary mitigation methods.

The experimental and theoretical analysis of the paper is solid.

**Weaknesses:**

There are no obvious weaknesses in the paper.

**Questions:**

I have a few questions for the authors：
1. Whether the “context-parametric inversion” phenomenon can be mitigated by using a smaller learning rate.
2. Would a parameter-efficient fine-tuning method like LoRA have the same problem?
3. Since the curve goes up and then down, does this mean that it is better to use a small amount of fine-tuning data rather than a large amount of fine-tuning data to enhance the context-following ability of the model.

---

> ### Author Response · Authors · 2024-11-17
> **Rebuttal by Authors**
>
> Dear Reviewer DZJy,
>
> Thank you for the feedback! We’re grateful for your acknowledgement of the value of our work. We address your comments below.
> \$ $
> ### **Effect of learning rate**
> > Whether the “context-parametric inversion” phenomenon can be mitigated by using a smaller learning rate.
>
> Thanks for the great question! No, context-parametric inversion cannot be mitigated by early stopping or using a smaller learning rate. This is because the drop in counterfactual performance during instruction tuning tends to happen *before performance on standard benchmarks converges*. As a result, addressing the problem by using a smaller LR or early-stopping may be undesirable.  For example, in Appendix A2, we have added a comparison in trends between LR 1e-4 and LR 1e-5, when finetuning Llama2-7B on Alpaca for the  same number of epochs. We continue to observe a drop in counterfactual performance even with a smaller LR. Note that while the magnitude of drop might seem smaller, it is because the training hasn’t converged with smaller LR (as highlighted by the low standard benchmark performance).
> \$ $
> ### **Would a parameter-efficient fine-tuning method like LoRA have the same problem?**
>
> Thanks for the great question! We observe context parametric inversion in both LoRA and full finetuning settings. Appendix A3 has results for LoRA (this is the default setting in the main paper for computational concerns) and Appendix A6 has results with full finetuning.
> \$ $
> ### **Since the curve goes up and then down, does this mean that it is better to use a small amount of fine-tuning data rather than a large amount of fine-tuning data to enhance the context-following ability of the model?**
>
> Thank you for the interesting question! Our theoretical analysis over a 1-layer transformer (Theorem 1) tells us the following insight. Rather than the dataset size, we have to think about the *composition of  types* of examples that are present in the training data and to what degree each type may (or may not) promote context-following ability.  In particular, we found that instruction tuning datasets contain many *non-context critical points* where a context is provided but the model can correctly respond to a user query using *either* the context or its parametric memory. For example, Alpaca contains reading comprehension questions about a true historical figure, and the model may already know the answer to this question from knowledge accumulated during pretraining. Furthermore, there are also some subject-critical examples which necessitates models to perform fact recall, thus strictly encouraging reliance on parametric memory. Our theorem requires that a nontrivial *percentage* of such training examples (non-context critical and subject-critical) must be present in the training data to observe the behavior of counterfactual performance peaking then falling during instruction tuning. Then, *regardless of the finetuning dataset size*, the drop in context-following ability would persist!
> \
> \
> Thanks again. Please let us know if you have any other questions!

---

### Official Review · Reviewer_9rEs · 2024-11-03

**Soundness:** 2
**Presentation:** 3
**Contribution:** 2
**Rating:** 8
**Confidence:** 4

**Summary:**

The paper investigates a phenomenon termed context-parametric inversion in large language models (LLMs), where LLMs' instruction following ability reduced after fine-tuned on a certain amount of instruction data despite a increase at the beginning. This study reveals that while fine-tuning initially increases the model's reliance on input context, it eventually shifts back toward using its internal, parametric knowledge, leading to context-related errors or hallucinations.

**Strengths:**

By examining multiple model families (Llama, Mistral, Pythia) and instruction datasets (TULU, Alpaca, UltraChat), the authors provide robust evidence for the findings in the paper.
This paper offers a theoretical framework explaining how gradients from context-critical and non-context-critical points interact during fine-tuning. This framework gives a deeper insight into why the inversion happens, moving beyond empirical results to provide a conceptual basis for future research.

**Weaknesses:**

Inconsistent x-axis in the figures, some of them do not start from 0 and lack of explanation on these x-axis. Inconsistent or unexplained x-axis scales can indeed impact the interpretability and robustness of the findings in the paper.

Typo in line 311, it should be Figure 3c instead of Figure 3b.

Missing reference: Instruction-following Evaluation through Verbalizer Manipulation. Li, S., Yan, J., Wang, H., Tang, Z., Ren, X., Srinivasan, V., & Jin, H. In 2024 Annual Conference of the North American Chapter of the Association for Computational Linguistics, 2024.

**Questions:**

Can you give a detail explain on the x-axis for each figure?

---

> ### Author Response · Authors · 2024-11-17
> **Rebuttal by Authors**
>
> Dear Reviewer 9rEs,
>
> Thank you for the feedback! We address your comments below.
> \$ $
> ### **Interpretability of X Axis in Our Plots**
> Thanks! We have added more detail in the figure captions in the updated manuscript. We were uncertain where the comment “some of the x axis do not start from 0” referred to, but do our best to address this concern here. To clarify, we do instruction tuning over pretrained LLMs (e.g., LLaMA-2-7b-base) which already have non-zero performance on various benchmarks. Thus the plots are not supposed to start from 0. If there is a particular figure that was unclear to you, please let us know! We are happy to resolve any other concern you may have or if we misunderstood what you were referring to here.
> \$ $
> ###  **Typo in line 311, it should be Figure 3c instead of Figure 3b.**
> Thank you! We have fixed this error accordingly.
> \$ $
> ### **Cite “Instruction-following Evaluation through Verbalizer Manipulation” Paper**
> Thank you for pointing out this missing work in our references! We have added this citation. Similar to other works like [1,2], this work identified scenarios where instruction tuned models have high variance in their performance on “unnatural” instructions.
>
> In contrast to these works, we focus on carefully studying the dynamics of instruction tuning (e.g. the data composition and optimization choices). We isolate the cause of the drop in context dependency and instruction following during instruction tuning. In particular, we identify the presence of *non-context critical points* in instruction datasets where a context is provided but the model can correctly respond to a user query using either the context or its parametric memory.  We provide a theoretical analysis of how the dynamics of instruction finetuning is affected by the model’s parametric knowledge and how this may stray models away from the intended instruction/context following behavior. We hope our analysis provides a deeper understanding of why such problems tend to occur in language models and how to fix them.
> \
> \
> We hope this addresses your concerns! Thanks again, please let us know if you have any other feedback.
> \$ $
> 1. Ian R. McKenzie, Alexander Lyzhov, et al. Inverse scaling: When bigger isn’t better, 2024.
> 2. Suhas Kotha, Jacob Mitchell Springer, and Aditi Raghunathan. Understanding catastrophic for-
> getting in language models via implicit inference, 2024.

---

> > ### Comment · Reviewer_9rEs · 2024-11-26
> >
> > Thanks for the clarification. It addressed my concerns, I decide to increase my rating to 8.

---

> ### Author Response · Authors · 2024-11-23
> **Hope to hear back soon**
>
> Dear Reviewer 9rEs,
>
> In our response above, we have tried to address all your questions and comments. To summarize, we have clarified the issue of interpretability of our x-axis and added the missing citation.
>
> We once again thank you for your time spent reviewing our work and hope to hear back from you soon. Please let us know if you have any additional questions!

---

### Official Review · Reviewer_knki · 2024-11-04

**Soundness:** 3
**Presentation:** 3
**Contribution:** 3
**Rating:** 8
**Confidence:** 3

**Summary:**

This paper studies an interesting phenomenon during instruction fine-tuning, in which context reliance first increases but later decreases as the model increasingly leverages parametric knowledge. Using three knowledge-conflict datasets to measure counterfactual accuracy and parametric accuracy during instruction finetuning, the authors demonstrate this phenomenon experimentally, across a few combinations of LLMs and instruction-tuning datasets. The paper includes a theoretical argument that aligns with the observation that removing data points where the context aligns with the parametric knowledge mitigates this phenomenon. Finally, the authors explore several strategies for mitigating this context-parametric inversion.

**Strengths:**

- The authors conduct a series of experiments that (1) demonstrate this context-parametric inversion is a persistent phenomenon, (2) test (and rule out) various hypotheses, (3) show that non-context-critical data points (data points where the context aligns with the parametric knowledge) are likely to blame, and (4) reveal the shortcomings of existing data augmentation or training approaches. These experiments are comprehensive and intuitively presented.

- The theoretical argument and justification for why this phenomenon occurs are reasonably easy to follow and sufficiently rigorous, certainly helping to validate the experimental results presented in Fig. 3(c).

**Weaknesses:**

- A few sections could use a bit of refinement for clarity. For instance, additional discussion on how counterfactual and parametric accuracy are measured on the context-parametric conflict datasets would be helpful beyond what is present at the beginning of Sec. 3. Another topic that could use a bit more explanation is the context-based filtering of Alpaca.

- Additional analysis on counterfactual data augmentation would be nice to have. The difference in results between Alpaca and TULU suggests that the ratio of counterfactual data included in the fine-tuning data mix is somewhat impactful. Additional ablations on this ratio would be interesting.

- I’m slightly concerned with the quality of the constructed context-parametric conflict datasets, given that most of the experimental results center around these datasets. For instance, for the CF_BIO task, the authors apply entity substitutions algorithmically rather than using an entity-substitution model as previous works have done in order to avoid “an incoherent context and an inaccurate estimate of the context-reliance”. Yet, looking at the provided examples reveals these inconsistencies are still present. For example, in the second CF_BIO example on pg. 19 line 993, “William Shakespeare” was correctly substituted with “Julius Caesar” at the beginning of the context, but later occurrences (particularly of just the last name “Shakespeare”) were left unchanged (see lines 997 and 1001). When measuring counterfactual accuracy, the context for the question “What is the name of the author who wrote Hamlet, Romeo and Juliet, Macbeth?” should not include the phrase “Shakespeare’s big break came with the success of Romeo and Juliet”.

**Questions:**

- Could you verify that the inconsistencies in the constructed context-parametric conflict datasets (see weaknesses above) are isolated incidents not present in the vast majority of the instances? Specifically for the CF_BIO task, was substitution only done for full entity names or did you make some attempt to catch analogous or shortened entity names?

- In Sec. 4.1 line 292, what do you mean by test set? The three context-parametric conflict datasets? If so, maybe another term besides “test set” can be used since “test set” implies that it’s drawn from distribution as the training set, which would not be the case between these context-parametric conflict datasets and instruction-tuning datasets.

- Related to Sec 4.2, can you expand on the filtering done on the Alpaca dataset to produce the context-only Alpaca dataset? How do you decide if an instance has “some input context”? Likewise, in Sec 4.3, is the perplexity loss computed using the base model (e.g., Llama-7b for Alpaca-7b)?

- Can you explain the setting in Fig. 4(d)? The title seems to indicate that it’s comparing your standard finetuning configuration to the QK finetuning configuration discussed in Sec. 6, but the legend indicates that those results only use the QK finetuning configuration on different datasets (Alpaca vs Alpaca context-only).

- In figs 5, 6, 7 in A.1 and figs 8 and 9 in A.2, is “ID Accuracy” the same as “Standard Benchmarks Performance,” or is there some nuanced difference that I’m missing?

- Are the training runs done with a fixed learning rate? Could decaying the learning rate during training diminish the prevalence of the context-parametric inversion by scaling down the large gradients from the non-context-critical data points?

Other comments not factored into my decision assessment:
- Sec A.4, line 903 (Fig 12 caption): “fullfinetuning” -> “full finetuning”

I am willing to raise my score if my questions/concerns are adequately addressed.

---

> ### Author Response · Authors · 2024-11-17
> **Rebuttal by Authors Part 1**
>
> Dear reviewer knki,
>
> We really appreciate your thoughtful feedback! We’re grateful for your acknowledgement of the value of our work. We address your concerns below:
> \$ $
> ### **Incorrect substitutions in CF_BIO**
> > 1. Could you verify that the inconsistencies in the constructed context-parametric conflict datasets (see weaknesses above) are isolated incidents not present in the vast majority of the instances? Specifically for the CF_BIO task, was substitution only done for full entity names...
>
> > 2. I’m slightly concerned with the quality of the constructed context-parametric conflict datasets, given that most of the experimental results center around these datasets…
>
> Thank you for catching this critical error in our CF_BIO dataset. Unfortunately, our code for entity substitution had not properly replaced some partial names due to a rendering error for apostrophes. We have addressed this problem in our CF_BIO v2.0 dataset and re-evaluated all our models. For transparency, we release all the datasets in our benchmark as a part of our supplementary materials. Fortunately, this problem is *isolated to CF_BIO alone* as other datasets do not require any entity-substitution. We’ve further manually checked that there is no occurrence of the context pointing to two different answers in every dataset.
>
> You can see Section A.1 of our Appendix for re-evaluated performances of all the instruction fine-tuned models. Due to the fix, all biographies now correctly refer to the substituted entity, so the zero-shot counterfactual performance improves generally for all base models. But critically, *context-parametric inversion persists* – we still see the trend of counterfactual performance peaking in the middle of instruction tuning, then dropping back down. This tells us that this phenomenon is not a trivial result of the error in our dataset. In fact, in all of our datasets (now the context clearly points to one counterfactual answer), we see that models, during fine-tuning, fall trap to a performance drop.
> \$ $
> ### **Related to Sec 4.2, can you expand on the filtering done on the Alpaca dataset to produce the context-only Alpaca dataset?**
>
> Thank you for the feedback! The filtration is easy to do for Alpaca as any example with a “context” is already tagged in the dataset. To be more specific, the Alpaca SFT dataset (tatsu-lab/alpaca on Huggingface) consists of 3 columns: “instruction,” “input,” and “output.” The “instruction” corresponds to the user instruction and the “input” is any additional context that is paired with the instruction. For example, an example could be “Instruction: Who won the marathon? ### Input: [Scoreboard]”. However, there are some pure fact-recall examples with a blank input, such as “Instruction: What are the three primary colors?” We filter out any examples with a blank “input.” We add this discussion to Section 4.2.
> \$ $
> ### **Likewise, in Sec 4.3, is the perplexity loss computed using the base model (e.g., Llama-7b for Alpaca-7b)?**
>
> Thank you for the feedback! Yes, we filter out examples based on the perplexity of the loss computed over the base model. We’ve made this change in 4.3. The goal of this experiment was to identify examples where the model is able to correctly respond using knowledge acquired from pretraining, regardless of whether a context is present. We found that these examples can negatively impact counterfactual performance. Admittedly, filtering these examples alone may not entirely resolve the drop. Theoretically, we showed that once the model *memorizes facts in the instruction tuning data*, even examples that were once context-critical can transition to becoming non-context-critical.
> \$ $
> ### **Counterfactual Data Augmentation**
> > The difference in results between Alpaca and TULU suggests that the ratio of counterfactual data included in the fine-tuning data mix is somewhat impactful. Additional ablations on this ratio would be interesting.
>
> Thanks for the feedback! Generally, we find in our work that even incorporating a small amount (2-5% of the total dataset) of entity-substituted data can partially mitigate the drop in counterfactual benchmarks such as CF_BIO, while on stylistically very different benchmarks such as CF_QUOTES, we do not see any gain. We agree a better general understanding of how much counterfactual data must be incorporated during instruction tuning is an important direction. To rigorously address this would require a large variety of CF data and training many models at different augmentation proportions.  Due to time and compute  constraints, we will leave this for future work, but we are happy to make further adjustments if this remains a concern!

---

> > ### Author Response · Authors · 2024-11-17
> > **Rebuttal by Authors Part 2**
> >
> > ### **Can you explain the setting in Fig. 4(d)?**
> > > The title seems to indicate that it’s comparing your standard finetuning configuration to the QK finetuning configuration discussed in Sec. 6, but the legend indicates that those results only use the QK finetuning configuration on different datasets (Alpaca vs Alpaca context-only).
> >
> > We apologize for the confusion! There was a typo in our legend. Blue denotes All Parameter finetuning and red denotes QK finetuning. Importantly, observe that the counterfactual performance drop is not as severe with QK finetuning. For example, an All Parameter fintuned model with 52% accuracy on standard benchmarks achieves 50% on CF_World_Facts, whereas a QK finetuned model with the same standard benchmark accuracy achieves a counterfactual performance of 65%. We have fixed figure 4(d) accordingly.
> > \$ $
> > ### **Does scaling down learning rate fix the problem?**
> > > Are the training runs done with a fixed learning rate? Could decaying the learning rate during training diminish the prevalence of the context-parametric inversion by scaling down the large gradients from the non-context-critical data points?
> >
> > Thanks for the question! In all of our experiments, we employ a cosine or linear learning rate decay. Regardless, we see a drop in performance. This is to say that in general,  standard learning rate scheduling is not enough to prevent the counterfactual performance drop. But you are right – hypothetically, doing extremely strong learning rate decay or *early-stopping* would indeed prevent the drop from happening. In Appendix A2, we have added a comparison in trends between LR 1e-4 and LR 1e-5, when finetuning Llama2-7B on Alpaca for the  same number of epochs. We continue to observe a drop in counterfactual performance even with a smaller LR. Note that while the magnitude of drop might seem smaller, it is because the training hasn’t converged with smaller LR (i.e. achieves lower standard benchmark performance). In general, addressing context-parametric inversion by learning rate decay or early-stopping may be undesirable since, as we highlight in our work, the drop in counterfactual performance during instruction tuning tends to happen *before performance on standard benchmarks converges*.
> > \$ $
> > ### **Additional discussion on how counterfactual and parametric accuracy are measured on the context-parametric conflict datasets**
> >
> > Thanks for the feedback! The counterfactual benchmarks are simple to evaluate over, as every example requires short-form responses of at most 2-3 words. For example, the answer to a question CF_Capitals, "What is the capital of France?", is either the counterfactual capital described in the context (“Lyon”) or the true capital (“Paris”). We evaluate the performance of models by sampling multiple responses from the model and measuring whether the factual or counterfactual answer is entailed in the model response. In particular, ``counterfactual accuracy'' measures whether the context-based answer is present in the model’s generation. Similarly, "parametric accuracy" measures whether the factual answer is present in the generation.  Let us know if this answers your question. We have clarified these points accordingly in the introductory paragraph in Section 3.
> > \$ $
> > ### **Other Clarification Points**
> > > In figs 5, 6, 7 in A.1 and figs 8 and 9 in A.2, is “ID Accuracy” the same as “Standard Benchmarks Performance,” or is there some nuanced difference that I’m missing?
> >
> > We apologize for the confusion! ID Accuracy means the same as Standard Benchmarks Performance. We have made this change to all the figures in the Appendix.
> >
> > > In Sec. 4.1 line 292, what do you mean by test set? The three context-parametric conflict datasets? If so, maybe another term besides “test set” can be used….
> >
> > We again apologize for the confusion! Yes, we meant the three context-parametric conflict datasets by the term “test set”. We have fixed this in the updated manuscript.
> > \
> > \
> > Thanks again for your thoughtful review. We hope this addresses your concerns!

---

> > > ### Comment · Reviewer_knki · 2024-11-17
> > >
> > > Thank you for taking the time to respond to my comments and for providing a revised manuscript. I appreciate the detailed efforts you have made to address my concerns. I will update my initial review accordingly.

---

### Official Review · Reviewer_BBKt · 2024-11-05

**Soundness:** 3
**Presentation:** 4
**Contribution:** 3
**Rating:** 8
**Confidence:** 4

**Summary:**

This work explores the context-reliance failure in instruction tuning, that is observed during the instruction finetuning of large language models. While finetuning is expected to improve a model's adherence to input context, the study finds that context reliance decreases as the training goes. The authors examine this behavior across several datasets (TULU, Alpaca, Ultrachat) and model families (Llama, Mistral, Pythia), and conduct comprehensive controlled studies to isolate the causes. The paper further provides theoretical analysis and suggests potential mitigation strategies.

**Strengths:**

- The identification and analysis of the context-parametric inversion phenomenon could contribute to our understanding of LLM behavior during instruction tuning. The findings in this work are helpful for future work in this direction.

- The analysis is conducted across multiple datasets and model families, ensuring the robustness of the findings. The paper includes controlled studies to rule out simple hypotheses, contributing to a deeper understanding of the phenomenon.

- The theoretical analysis provides a solid foundation for understanding the observed behavior and suggests potential mitigation strategies.

**Weaknesses:**

The learning rate also has a non-trivial impact on performance. What are the considerations for selecting 1e-4 and 1e-5 as learning rates? What are the trends across different learning rates?

**Questions:**

Typo: Line 311 seems to need changing to Figure 3c?

---

> ### Author Response · Authors · 2024-11-17
> **Rebuttal by Authors**
>
> Dear Reviewer BBKt,
>
> Thank you for the feedback! We’re grateful for your acknowledgement of the value of our work. We address your comments below.
> \$ $
> ### **How does learning rate affect the drop?**
> > The learning rate also has a non-trivial impact on performance. What are the considerations for selecting 1e-4 and 1e-5 as learning rates? What are the trends across different learning rates?
>
> We select the learning rate that gives the best performance on the standard benchmarks (GSM8k, MMLU, Arc_challenge and SQuAD). However, we do observe context parametric inversion even when using a smaller learning rate (LR). In Appendix A2, we have added a comparison in trends between LR 1e-4 and LR 1e-5, when finetuning Llama2-7B on Alpaca for the  same number of epochs. We continue to observe a drop in counterfactual performance even with a smaller LR. Note that while the magnitude of drop might seem smaller, it is because the training hasn’t converged with smaller LR (as highlighted by the low standard benchmark performance). In general, context-parametric inversion cannot be resolved simply by using classical fixes like early stopping or smaller LR. What we highlight in our work is that the drop in counterfactual performance during instruction tuning tends to happen *before performance on standard benchmarks converges*. As a result, addressing the problem by learning rate decay or early-stopping may be undesirable.
> \$ $
> ### **Typo: Line 311 seems to need changing to Figure 3c?**
> Thank you for catching this error! We have made this fix in our submission.
> \
> \
> We thank the reviewer again for their time and consideration. We are more than happy to address any other concern they might have.

---

> > ### Comment · Reviewer_BBKt · 2024-11-25
> >
> > Thank you for addressing my concerns. I will keep my original rating.

---

### Meta-Review · Area_Chair_35w8 · 2024-12-22

**Metareview:**

Based on the reviews, I recommend accepting this paper. The work identifies and thoroughly analyzes an important phenomenon called "context-parametric inversion" in instruction-tuned large language models, where models counter-intuitively become less reliant on input context as training progresses. The study provides robust evidence by examining multiple model families and instruction datasets, supported by comprehensive controlled studies and theoretical analysis. The theoretical framework explaining the gradient interactions during fine-tuning provides valuable insights for future research. The identification of this phenomenon and proposed mitigation strategies make this a valuable contribution to our understanding of LLM behavior during instruction tuning.

**Additional Comments On Reviewer Discussion:**

I have read the messages in the discussion period and my opinion has been summarized as in the metareview above. I considered these points in my recommendation.

---

### Decision · Program_Chairs · 2025-01-22

Accept (Oral)